# Harnessing droplet microfluidics and morphology-based deep learning for the label-free study of polymicrobial-phage interactions

Anuj Tiwari [1], An Mei Daniels [1,2], Remy Chait[1,2,3], Robyn Manley [3] ✉ & Fabrice Gielen [1,4] ✉

Evaluating the impact of bacteriophages on bacterial communities is required to assess the future utility of phage therapy. Methods able to study bacterial polycultures in the presence of phages are useful to mimic evolutionary pressures found in natural environments and recapitulate complex ecological contexts. Bacteriophages can drive rapid genetic and phenotypic changes in host cells. However, the presence of other bacteria can also impact bacterial densities and community structure and classical methods remain lengthy and resource intensive. Here we introduce a microdroplet-based encapsulation method in which bacterial co-cultures are imaged using Z-stack brightfield microscopy. The method relies on automated droplet imaging using an AI-based autofocus function, coupled with morphology-based deep learning models for accurate identification of two morphologically distinct bacterial species. We monitor the interactions between bacterial mono- or co-cultures of *P. aeruginosa* and *S. aureus* in the presence of a *P. aeruginosa* phage growing in 11 picolitre droplets for up to 20 h. We demonstrate quantification of growth rates, bacterial densities and lysis dynamics of the two species without the need for plating. We show that a potent lytic phage of *P. aeruginosa* can keep its density low long-term when in the presence of *S. aureus*.

Bacteriophages are abundant in nature and major drivers of bacterial evolution. The use of phages to lyse specific bacteria has recently seen a resurgence in interest as a promising approach for tackling antibiotics resistance and diseases beyond infections. In addition to potential live drugs for human treatment, phages can find applications in decontamination of crops, animal feeds or hospital settings. However, bacterial phenotypes triggered by phage exposure in monocultures can significantly differ from those found in polymicrobial communities[1]. For example, rapid resistance acquisition and high fitness of resistant bacteria have often been reported when using monocultures. Yet, it is known that this evolutionary outcome of bacteria-phage interactions can be fundamentally altered by the presence of a polymicrobial community and this could be a key consideration in the development of future phage-based therapies[2]. It is increasingly clear that resistance mechanisms are context dependent and that phage resistance arise differently in vitro and in natural environments. For example, surface modifications and receptor shedding have been reported in laboratory conditions while CRISPR-Cas based defense systems are seen more often in polymicrobial communities due to fitness costs of certain types of resistance[3]. Beyond how resistance eventually arise, the question is whether microbial diversity promote or impede phage infections and how polymicrobial-phage interactions shape community structure.

In a study by Testa et al., a sensitive *P. aeruginosa* strain PAO1 was cultured with its lytic phage in the presence of an insensitive *P. aeruginosa* strain PA14. It was found that inter-strain competition reduced resistance evolution in the susceptible strain[4]. In another study by Harcombe et al. with *E. coli* B and *Salmonella enterica*, T5 and T7 phages kept *E. coli* density levels low in the two-species cultures although densities returned to the same levels as without phages in *E. coli* monocultures[5]. It was recently shown that phages are most likely to affect community composition when encounter

[1]Living Systems Institute, Faculty of Health and Life Sciences, University of Exeter, Exeter, UK. [2]Natural Sciences, Department of Physics and Astronomy, Faculty of Environment, Science and Economy, University of Exeter, Exeter, UK. [3]Department of Biosciences, Faculty of Health and Life Sciences, University of Exeter, Exeter, UK. [4]Department of Physics and Astronomy, Faculty of Environment, Science and Economy, University of Exeter, Exeter, UK. ✉e-mail: r.manley@exeter.ac.uk; f.gielen@exeter.ac.uk

rates with susceptible hosts is high, over short timescales, and in relatively simple communities[6].

We chose to co-culture of *P. aeruginosa* (PA14 Δ*flgK*, referred to as PA) and *S. aureus* (MSSA476, referred to as SA) as a model two-species community. PA is a common cause of nosocomial infections, accounting for 10–20% of infections in most hospitals and the combination with SA has high relevance to cystic fibrosis lung environments and chronic wound infections[7]. Many reports found that PA strongly outcompetes SA in liquid and biofilm cultures[8,9]. This is in part thanks to the secretion of several virulence and quorum sensing compounds including pyocyanins and polyphosphates which disrupts *S. aureus* respiratory processes and can cause oxidative stress[10]. However, other reports found culturing media conditions that support *P. aeruginosa* and *S. aureus* coexistence in vitro[11]. *S aureus* growth within biofilms has also been shown to display high dependency on oxygen availability[12].

Current methods for studying polymicrobial cultures are lengthy and labour intensive, typically based on liquid cultures followed by quantification of the relative abundance of either bacterial species found by plating on suitable selective agar plates. Standard optical density measurements cannot report on the relative growth of multiple species in the same mixed culture[13]. Other methods include flow cytometry of fluorescently labelled cells, or molecular techniques such as targeted sequencing, multiplex qPCR-based methods[14]. For example, Kehe et al. used fluorescent-based color codes and assays to study polymicrobial interactions at large scale but only at population level[15].

Recent innovations in microfluidics have enabled powerful approaches to study microbial communities. Within single phase devices, the mother machine devices have been extensively used. For instance, Dal Co et al. have advanced the field with devices to study ecological interactions and feedback between environmental structure and microbial behaviour, though most systems rely on flow-through or 2D culturing formats[16]. Gralka et al. have focused on spatial dynamics and stochastic effects in bacterial colonies using planar microfluidic chambers and agarose-based devices, highlighting genetic drift and sectoring phenomena[17]. Early droplet-based co-cultivation systems such as that of Herrera-Estrella et al. demonstrated the utility of fluorescently labelled bacteria for high-throughput mapping of microbial interactions in droplets, highlighting the potential of droplet platforms for studying microbial consortia[18].

Imaging methods can be used to identify individual bacteria and detect specific phenotypes[19]. However, most of the current imaging methods require cells to grow in single layers with limited relevance to 3D bacterial niches[20,21]. Studies have shown that formats (cell layers, biofilms, planktonic) in which bacterial cells grow influence their gene expression and in turn their response to antimicrobials. In a seminal study, Hoiby and colleagues showed that the formation of *P. aeruginosa* inhibition zone during tobramycin agar diffusion susceptibility tests is due to a switch from planktonic growing bacteria to the biofilm mode of growth[22]. Recently, Ramachandran et al. showed that planktonic cells exposed to shear flow have altered transcriptomics profiles[23]. The study of the link between cell motility and virulence is also important as in clinic contexts including respiratory mucus environments (e.g., cystic fibrosis), and wound exudates[24].

Despite the relevance of the liquid assay format, counting and identifying planktonic cells in 3D environments is more challenging due to the spreading and random orientation of bacteria at any time. Microfluidic encapsulation methods enable facile trapping of individual cells necessary for long-term imaging and rapid generation of co-cultures with precise control over cellular encapsulation parameters (e.g. volumes, trapping locations) and the creation of diverse microbial communities with starting bacteria numbers following Poisson statistics[25,26]. Microbial interactions have been studied in droplets, for instance with the co-encapsulation of symbiotic fluorescent bacteria[18]. These studies have required fluorescent reporters and relied on quantifying overall fluorescent levels to estimate bacterial densities over time[27,28]. Other microfluidic formats to study polymicrobial cultures listed in Supplementary Table 1 either lack single-cell resolution or require fluorescent labelling.

We have previously demonstrated label-free counting of *E. coli* cells trapped in non-contacting static microdroplets with a low-throughput approach that enabled tracking of individual bacterial division events[29]. Here we expand the method to two-species cultures using rod-shaped and cocci-shaped cells trapped within shallow water-in-oil microdroplets in which strains interact. We show that deep learning object detectors can accurately count the number of cells of both species based on their morphological signatures[30]. A Z-stacking imaging method enables accurate counting across the 3D bacterial niches. We further introduce an AI-based autofocus function enabling stable and reproducible focus over timescales of hours while the microscope stage and objective undergo repetitive XY and Z motion respectively. Previous studies have investigated the use of autofocus methods using convolutional neural networks to determine the focal plane of an image in a single shot, and identify the distance and direction needed to return to focus[31]. This is more efficient than other implementations using multiple slices a known distance apart to extrapolate focal distance[32]. However, all of the networks were trained on stained or auto-fluorescent tissue samples, where all cells were in the same plane[33–35]. Our autofocus system was trained on 5 separate classes corresponding to 5 different focal range for the droplets and a feedback loop was implemented to converge towards best focus for every droplet screened. This enabled us to collect reproducible experimental data with non-fluorescent bacterial cells for long-term time-lapse imaging.

Using the platform, we monitored the relative growth dynamics of *P. aeruginosa* (PA) and *S. aureus* (SA) for up to 20 co-cultures per experiment growing in confined 12 picolitre droplets for up to 24 h. We challenged monocultures and polycultures to a lytic phage of *P. aeruginosa* and demonstrate quantification of growth rates, relative bacterial densities and lysis dynamics for the two species.

Our data show that in the absence of phage, SA populations in a mixed SA + PA community had usually, but not always, reduced growth rates compared to monoculture controls. The presence of PA phage P278 fully lysed or suppressed PA while promoting *S. aureus* growth in two-species cultures over timescales of over 5 h suggesting long-term growth suppression effect.

## Results

We aimed at establishing a high-throughput method for the quantitative analysis of bacterial co-cultures and application to phage infection dynamics. The droplet imaging system developed in this study integrates automated time-lapse image acquisition, focal plane correction using deep learning, and axial drift (XY axis) compensation using Hough transform analyses (Fig. 1). This system enables the precise imaging of individual droplets sequentially, with the droplet chip mounted on a moveable, programmable platform, ensuring that both the focal plane and droplet imaging conditions are maintained throughout long-duration experiments.

The automated microscope stage enabled acquisition of Z stacks for up to 20 droplets over times of up to 24 h, providing Z-stack time lapse imaging every 5 min. To correct for slight differences in focal planes for every droplet due to imperfect device flatness and possible vertical drift, we implemented a deep learning based autofocus method. Lateral drift was corrected by centering droplets using Hough transforms before every Z-stack imaging. The quasi-flat droplets (6 μm in height, 60 μm in diameter) decreased the need for large numbers of Z slices, significantly speeding up the overall image acquisition process while still leaving cells unconstrained.

### Deep learning for autofocus and bacterial cell detection

**Focus correction using image classification**. We implemented an autofocus method using YOLOv8 based image classification (Fig. 2). YOLOv8 is a version of the YOLO (You Only Look Once) real-time object detector which can perform tasks such as object detection and image classification. YOLOv8 was chosen due to its sophisticated data augmentation methods, which increases model robustness so it can be used on a wide range of imaging conditions, as well as its very high inference speed, which allows for very fast focal adjustments in-between

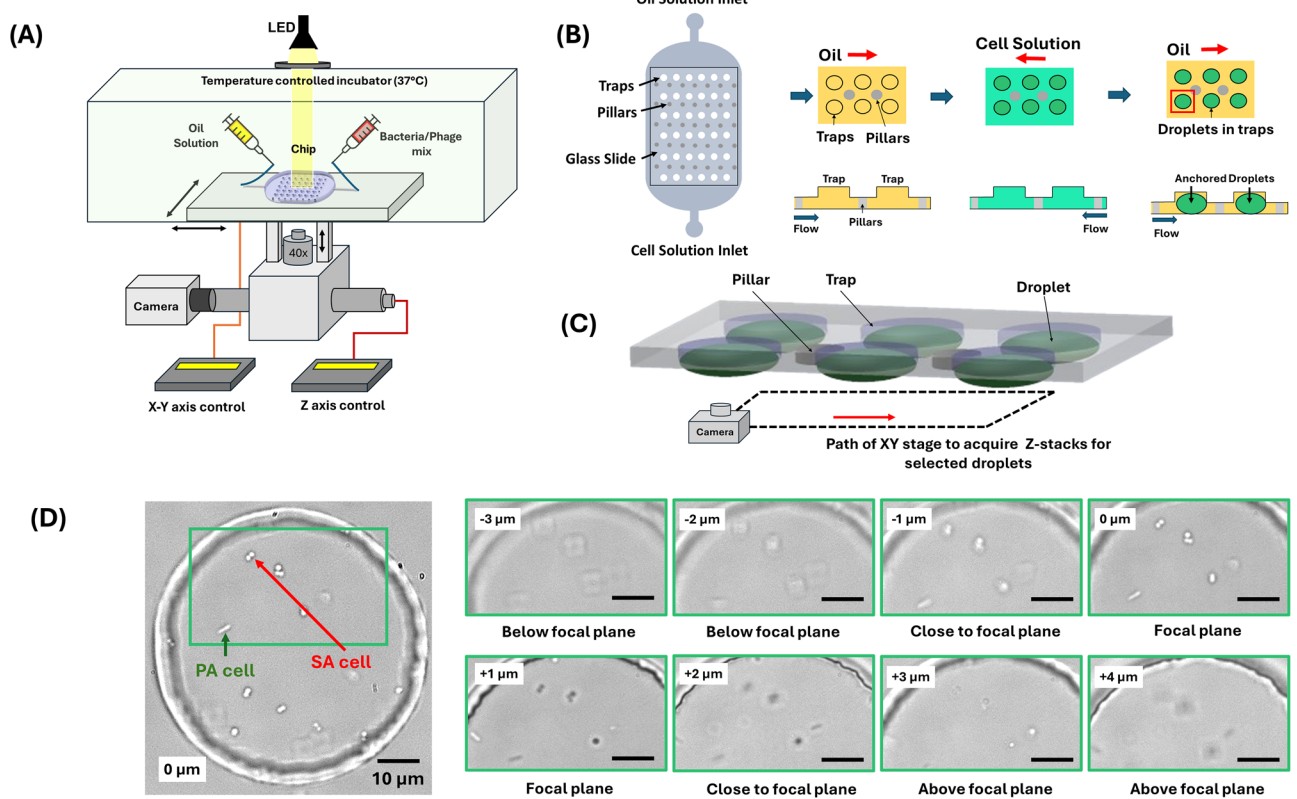

**Fig. 1 | Microfluidics and imaging setup to study bacterial growth and bacteriophage interactions in anchored droplets. A** Schematic of the microfluidic experimental setup, including an inverted microscope equipped with a 40x magnification lens, a temperature-controlled incubator (37° C), a microfluidic chip comprising oil and bacterial/phage solution inputs, and an integrated camera. **B** Illustration of the droplet generation process within the microfluidic chip, highlighting key stages of droplet formation using multiple flow directions, resulting in an anchored droplet array. **C** Automated imaging of multiple droplets using XY stage motion and Z-stacking to acquire imaging data to generate accurate counts. **D** High-resolution microscopy images of bacterial cells (PA and SA) encapsulated in droplets, with scale bars of 10 μm. The green box highlights an example region of interest. Sequential microscopy images of a droplet loaded with bacteria, taken at different Z-axis positions (from -3 μm to +4 μm), showcasing the 3D positioning and morphological features at different focal planes for each species of bacterial cells.

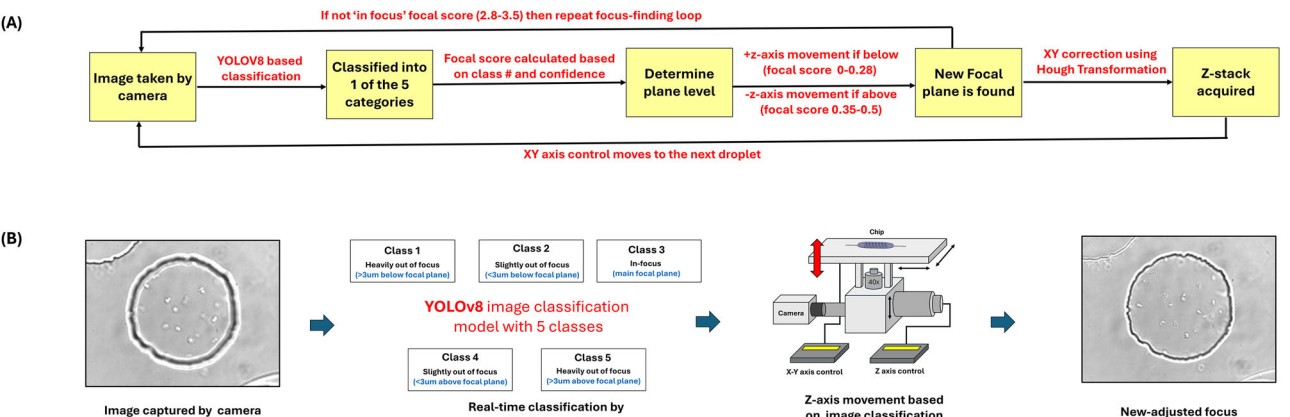

**Fig. 2 | Autofocus using YOLOv8-based deep learning. A** The workflow illustrates an automated focal adjustment system for droplet imaging using a YOLOv8-based classification model. Images are captured at 40x magnification and classified into five categories: heavily out of focus (>3 μm below or above the focal plane), slightly out of focus (<3 μm below or above the focal plane), and in-focus (close to the main focal plane). A focal score is calculated based on the classification and confidence level (see Eq. 1). An iterative focus adjustment loop is implemented if the score is not within the in-focus range (corresponding to a score of 2.8–3.5, with a score of 3 being the predicted best focus). The Z-axis is adjusted accordingly, with positive movement for below-focus scores (0–0.28) and negative movement for above-focus scores (0.35–0.5), until the correct focal plane is located. Subsequently, droplet centering in the XY plane is performed using Hough Transformation to identify circular traps, and a Z-stack of the droplet was acquired. **B** The bottom images illustrate the classification process, transitioning from an out-of-focus to an in-focus image. Supplementary Movie S1 shows the functioning of the focus correction.

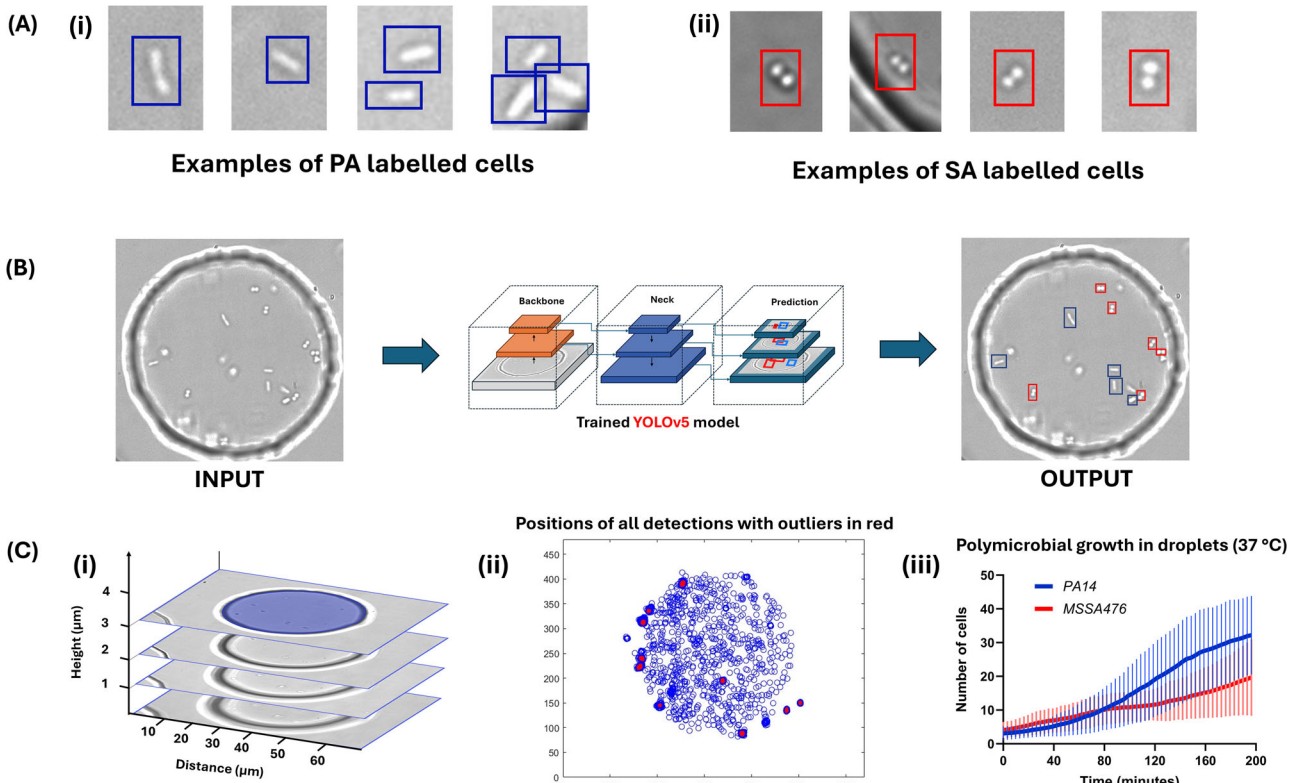

**Fig. 3 | Overall workflow from labelling cells to generating accurate counts. A** The examples of PA14 Δ*flgK* and MSSA476 morphologies as labelled for training the YOLOv5 object detection model (**A**-i) Examples of cells with rod shaped morphology labelled as PA14 Δ*flgK* (**A**-ii) Examples of cocci-shaped cells labelled as MSSA476. **B** Object detection using YOLOv5x. Z-stack images are processed using the trained models to obtain bounding boxes with classification and positions of cells as seen in the output image. The blue boxes are the model's prediction for PA14 Δ*flgK* cells and the red boxes are predictions for MSSA476 cells. **C** Generating cell counts. **C**-i Each Z-stack is treated as one time point and cell count in obtained by superimposing all bounding boxes. (**C**-ii) The blue dots represent the number of cells and their position detected during one experiment and the red dots represent the counts that were ignored due to clustering or false detections (x-axis is width in pixels and y-axis is height in pixels). **C**-iii Cell counts are obtained using moving average and plotted with standard deviation.

imaging. A custom training dataset of 3320 images was assembled from previous microfluidics experiments using circular traps and Z-stack imaging. Images of differing quality were chosen with varying brightness, background noise and artifacts, as well as different trap and droplet sizes, bacterial species to maximize the robustness and flexibility of the model. These images were then split into 5 classes – traps or droplets that appeared 'in focus', slightly below or above the plane (approximately <3μm out of focus based on the Z-stack) and greatly out of focus above or below the plane (>3μm). The images were randomly split into a training and validation dataset, with an 85:15 ratio. The YOLOv8-nano model, chosen for its small size and therefore high detection speeds, was then trained on the custom dataset, which showed very high accuracy (~96%) and low loss at the end of the training (refer to SI section 2 for more information on model training and accuracy parameters).

**Bacterial morphology detection using deep learning.** *P. aeruginosa* strain *PA14 ΔflgK* and *S. aureus* strain *MSSA476* were chosen for this study based on their marked morphological differences (Fig. 3A) and relevance to disease modelling (e.g., cystic fibrosis model). The Δ*flgK* mutants had low motility compared to wild type PA, and this simplified the data analysis given limited cell motion of the Δ*flgK* mutants during Z-stack imaging.

All the data was generated in the form of brightfield Z-stack images corresponding to individual time points. A typical stack had 40 images imaged every 0.5 μm across a height of 20 μm. Every image was analyzed separately using single-stage object detection based on deep learning (SI, section 10). YOLOv5x (extra-large version optimized for high accuracy) was the model of choice to perform the detection-based analysis. YOLOv5x

architecture is made up of 25 convolutional layers, and better detection accuracy compared to other models in the YOLOv5 family[27]. Three different models were trained for morphological detections for the analysis of three separate categories. The first model was trained to detect *P. aeruginosa* cells with rod-shaped morphology. The second model was trained to detect cocci shaped *S. aureus* cells with circular morphology. The third model was trained to detect both *P. aeruginosa* and *S. aureus* cells in a co-culture as two separate classes. The results of the model training are shown in SI section 2. We achieved a detection accuracy of ~99%, ~91% and ~97% for PA model (Supplementary Fig. 1), MSSA476 (Supplementary Fig. 2) model and bi-microbial model (Supplementary Fig. 3) respectively. The higher detection performance for PA compared to MSSA476 likely arises from differences in morphologies within each species: PA14 Δ*flgK* cells exhibit a well-defined, elongated rod-shaped morphology. In contrast, MSSA476 cells are cocci that often appear as singlets, diplococci, tetrads or grape-like aggregates, and can exhibit irregular or overlapping forms.

We generated anchored droplets using individual cell species PA14 Δ*flgK* and MSSA476 in LB media as well as their co-cultures. Tables 1 and 2 show all experimental conditions tested using our anchored droplet-based setup and the detection models used in all cases.

**Growth of individual bacterial species in anchored droplets.** The first step in validating our droplet co-culture methodology was to conduct growth experiments for individual species within droplets. For such experiments, we targeted a low initial mean cell number (e.g., 1–3 cells per droplet) by adjusting OD$_{600}$ accordingly. The time resolution was dependent on the total number of droplets screened and the number of images recorded per Z-stack. Typically, the platform would perform a

**Table 1 | Growth experiment conditions for individual and polymicrobial droplets**

| Experiment | OD$_{600}$ PA14 Δ*flgK* | CFU/mL- PA14 Δ*flgK* | OD$_{600}$ MSSA476 | CFU/mL-MSSA476 |
|---|---|---|---|---|
| PA14 Δ*flgK* Growth 1 | 0.11 | $6 \times 10^7$ | – | – |
| PA14 Δ*flgK* Growth 2 | 0.14 | $9 \times 10^7$ | – | – |
| PA14 Δ*flgK* Growth 3 | 0.14 | $9 \times 10^7$ | – | – |
| MSSA 476 Growth 1 | – | – | 0.14 | $1.3 \times 10^7$ |
| MSSA 476 Growth 2 | – | – | 0.13 | $1.1 \times 10^7$ |
| MSSA 476 Growth 3 | – | – | 0.14 | $1.2 \times 10^7$ |
| Polymicrobial Growth 1 | 0.10 | $4 \times 10^7$ | 0.16 | $1.5 \times 10^7$ |
| Polymicrobial Growth 2 | 0.31 | $2.25 \times 10^8$ | 0.40 | $0.5 \times 10^8$ |

**Table 2 | Lysis experiment conditions for individual and polymicrobial droplets**

| Experiment | CFU Count (PA14 Δ*flgK*)/mL | OD$_{600}$ PA14 Δ*flgK* | OD$_{600}$ MSSA476 | Phage titer concentration (Phage - P278) (PFU/mL) | Volume of cell culture (μL) | Volume of phage lysate (μL) | MOI |
|---|---|---|---|---|---|---|---|
| PA14 Δ*flgK* lysis 1 | $4.00 \times 10^8$ | 0.63 | – | $8 \times 10^9$ | 200 | 200 | 20 |
| PA14 Δ*flgK* lysis 2 | $6.00 \times 10^8$ | 0.77 | – | $8 \times 10^9$ | 200 | 10 | 0.6 |
| PA14 Δ*flgK* lysis 3 | $5.40 \times 10^8$ | 0.67 | – | $8 \times 10^9$ | 200 | 5 | 0.2 |
| MSSA476 + phage | – | – | 0.34 | $8 \times 10^9$ | 200 | 10 | – |
| Poly lysis 1 | $6.40 \times 10^8$ | 0.81 | 1.14 | $8 \times 10^9$ | 200 | 40 | 2.5 |
| Poly lysis 2 | $3.50 \times 10^8$ | 0.45 | 0.85 | $8 \times 10^9$ | 200 | 25 | 5 |

screen of 10 droplets in about 1 min. To correct for counting errors, we calculated a moving average of cell numbers obtained across a window of 8 consecutive time-points. Errors included missed detections close to the water-oil or trap boundary, difficulty in counting individual SA cells within aggregates. Static detections were removed as they were most likely false detections (see SI section 3, Supplementary Fig. 4). The PA14-v5 model built solely with images of PA14 Δ*flgK* cells was used to analyze experiments involving the PA14 Δ*flgK* cell strain including the droplet growth control as well as the interaction between PA14 Δ*flgK* cells and P278 phage. The MSSA-v5 model built with images of SA cells was used to analyze all MSSA476 related data including MSSA growth controls as well as the interaction between MSSA476 and P278 phage. The Bimicrobial-v5 model was used to analyze all co-culture experiments including co-culture growth and P278 interaction experiments.

**Growth of PA14 Δ*flgK* in anchored droplets at 37° C.** Experiments were conducted to analyze the growth of PA14 Δ*flgK* cells in droplets starting at varying bacterial densities as mentioned in Table 1. Assuming an ellipsoidal droplet of ~ 11 pL in volume, the initial mean number of cells in each droplet could be anticipated with Poisson statistics based on the loading optical density (OD$_{600}$) of the cell culture. Across different droplets, the data was analyzed using our cell counting method to calculate specific growth rates and cell doubling times as explained in methods. Figure 4A shows the change in cell count over time. For *P. aeruginosa* strain PA14 Δ*flgK*, the fastest cell doubling time obtained using our method was 26.52 min with an average cell doubling time of 33.96 ± 6.69 min calculated over all droplets observed and analyzed.

**Growth of MSSA476 in anchored droplets at 37° C.** For *S. aureus* strain MSSA476, the cell loading densities tested are mentioned in Table 1. The fastest cell doubling time observed was calculated as 41.48 min with an average cell doubling time of 69.49 ± 17.78 min. For one example droplet, Fig. 4B shows the change in cell number over time.

**Co-culture growth of PA14 Δ*flgK* and MSSA476 in anchored droplets at 37° C.** To generate droplets containing both PA14 Δ*flgK* and

MSSA476 cells, cells of each species in exponential growth phase were mixed in specific ratio as detailed in Table 1. A total of 12 droplets were analyzed across two experiments (Fig. 4C). For PA14 Δ*flgK* cells in a co-culture with MSSA476 cells, the fastest cell doubling rate was calculated to be 30.96 min with an average cell doubling time of 35.06 ± 4.86 min. For MSSA476 cells growing in a co-culture with PA14 Δ*flgK* cells, the fastest cell doubling rate was calculated at 43.39 min with an average cell doubling rate of 73.34 ± 21.07 min. Starting at similar cell numbers, PA14 Δ*flgK* cells in most cases become the dominant species and outgrow MSSA476 cells. Individual droplet cell numbers can be seen in SI section 4, Supplementary Fig. 5. For comparison, we performed bulk co-culture experiments of *P. aeruginosa* PA14 Δ*flgK* and *S. aureus* MSSA476 in the same LB medium. The data shows (SI section 5, Supplementary Fig. 6) that while PA14 Δ*flgK* grows comparably in both mono- and co-cultures, MSSA476 reaches significantly lower densities in polycultures, with its population reduced by over an order of magnitude after 24 h.

### Interaction of phage P278 with both species

To uncover bacteria-phage dynamics in a bi-species community, we tested a previously uncharacterized *P. aeruginosa* bacteriophage named P278 with both cell species individually and in co-cultures (Fig. 5). Phage P278 was isolated by screening 35 phages from the Exeter Phage Library against PA14 Δ*flgK*. The result of the screen is shown in SI section 6, Supplementary Fig. 7. Phage 278 was chosen for its good lytic efficiency against PA14 Δ*flgK* and absence of apparent impact on SA growth.

Phage life cycle parameters were initially characterized using the one step growth curve method[36]. The burst size (average number of phages released per infected cell) was calculated to be 45 and the time to first burst was calculated as 37 min (SI section 6, Supplementary Table 2 and Supplementary Figs. 8 and 9). Experiments were performed at various initial multiplicity of infection (MOI) to understand the relationship between phage loading and individual and polymicrobial growth dynamics. MOIs were calculated for every experiment as described in Table 2.

**Interactions between PA14 Δ*flgK* and phage P278 in anchored droplets.** Experiments were conducted at 3 different MOI conditions to

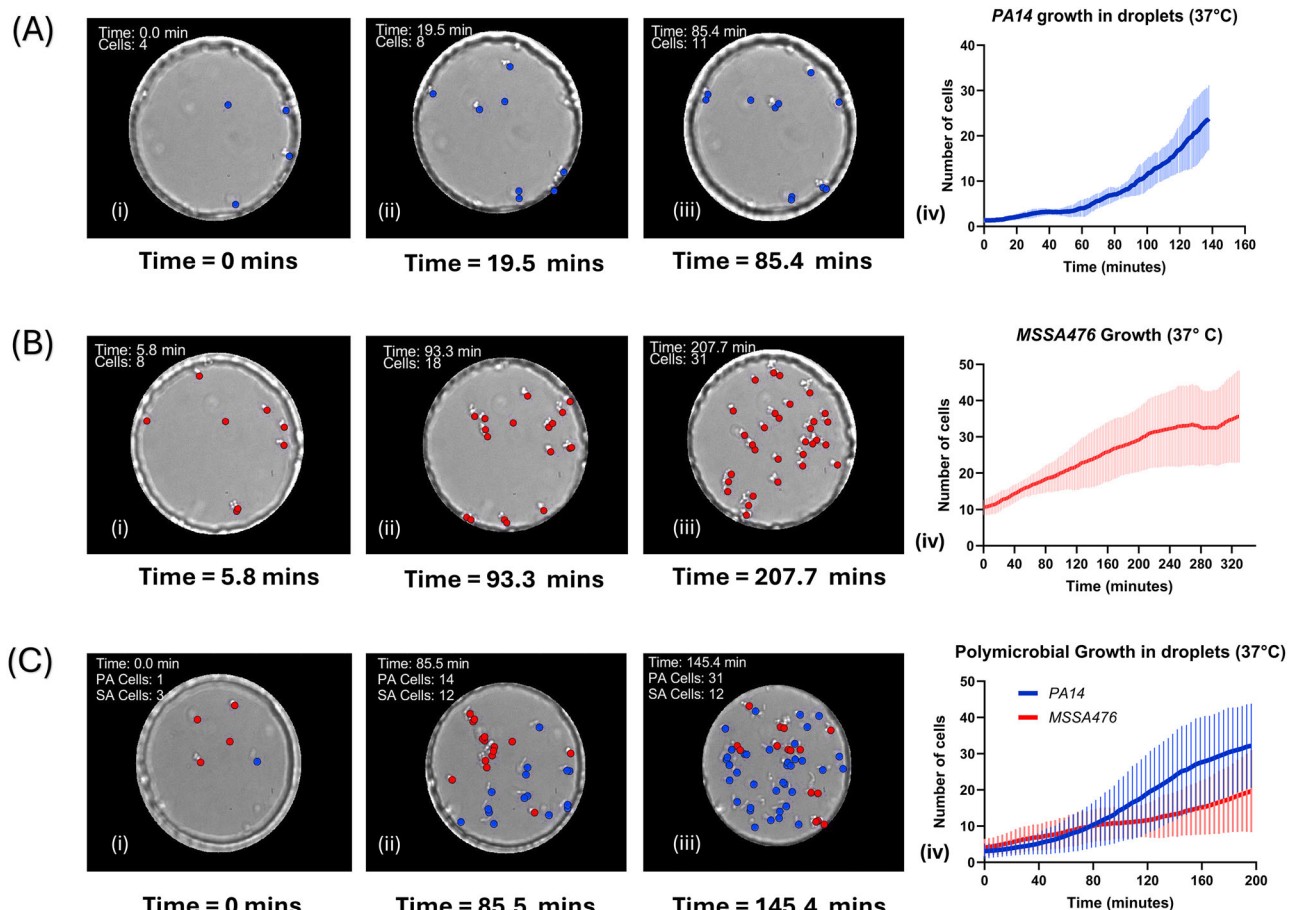

**Fig. 4 | Growth dynamics of PA14 Δ*flgK*, MSSA476, and their polymicrobial interactions in droplets. A** Growth of *P. aeruginosa* PA14 Δ*flgK* over time in droplets at 37° C. The line plot shows the mean number of cells (solid blue line) as a function of time, with the shaded area representing the standard deviation across growth experiments in droplets pooling the experiments shown in Table 1 (total number of droplets screened: 6 for PA14 Δ*flgK*, 6 for MSSA476, and 12 for co-cultures). Corresponding brightfield images depict snapshots of a selected droplet at various points, with individual detected cells marked by blue dots. **B** Growth of *S. aureus* MSSA476 over time in droplets at 37° C. The red line and shaded region represent the mean and standard deviation respectively. Brightfield images highlight cell growth within a droplet at different time points, marked by red dots. **C** Polymicrobial growth of *P. aeruginosa* PA14 Δ*flgK* and *S. aureus* MSSA476 in droplets at 37° C. The blue and red lines represent the mean cell counts of *P. aeruginosa* (PA) and *S. aureus* (SA), respectively, with shaded regions indicating standard deviations. Representative brightfield images illustrate the growth of a co-culture within a droplet at different time points, with *P. aeruginosa* cells marked by blue dots and *S. aureus* cells by red dots as seen in Supplementary Movie S2. All error bars represent one standard deviation from the mean.

observe the impact of lytic phage P278 on PA14 Δ*flgK* cell strain. In all phage experiments, we targeted an initial cell count of at least 10 cells from both species, such that they were still in exponential growth phase but were in sufficient numbers for quantifying lysis accurately.

The experimental conditions are listed in Table 2. Figure 5A represents the response of PA14 Δ*flgK* strain with an MOI of 0.6. A total number of 15 droplets were observed over 16 h. Cells were found to initially grow for the first 50 min before lysing afterwards. The population declined significantly over the duration of the experiment, reaching close to zero PA cells after 8 h. Figure 5B shows the lysis of PA14 Δ*flgK* cells at MOI of 20. A total of 8 droplets were observed for over 3 h. We observed a change in PA14 Δ*flgK* cell morphology - sufficient for cells not to be detected by the model - within 20 min, indicating faster cell lysis due to excessively high number of phage particles compared to the number of cells in each droplet. As seen in Fig. 5, even at high MOIs, we observed morphologically intact cells present at the end of the assays. However, the lack of growth long-term indicates that such cells, although displaying correct morphology, may not be viable.

To investigate low phage loads, we also conducted an experiment with an MOI of 0.2. This corresponded to 4–14 phages per droplet given to chosen bacterial densities. We imaged 12 droplets over 20 h and observed an initial increase in the number of cells followed by a decrease and stabilization in cell number over time. Figure 5C represents the dynamics of the cell-

phage interaction. Individual curves for all droplets are shown in SI section 7, Supplementary Fig. 10. In all phage experiments, we observed phage-induced lysis, indicating all droplets had sufficient number of phages to initiate infection cycles.

**Interactions between MSSA476 and phage P278 in anchored droplets.** We conducted control experiments to check if phage P278 displayed any interaction with MSSA476 cells in droplets. Figure 5D represents the growth observed when phage P278 was introduced to MSSA476 cells in an exponential growth phase. The red line represents the average growth, and the shaded region represents the standard deviation. A total of 6 droplets were observed across 3 experiments. The fastest cell doubling rate was calculated to be 44.39 min with an average cell doubling rate of 69.16 ± 16.03 min. A batch culture assay was performed in microtiter plate format to confirm MSSA476 was not affected by the presence of P278 (Supplementary Fig. 11).

**Interactions between co-cultures of PA14 Δ*flgK* and MSSA476 with phage P278 in anchored droplets.** The final set of experiments were conducted to study the impact of phage P278 on a co-culture of PA14 Δ*flgK* and MSSA476 cells. Two experiments were conducted at MOI of 2.5 and MOI of 5 respectively. These MOIs were chosen based on Fig. 5

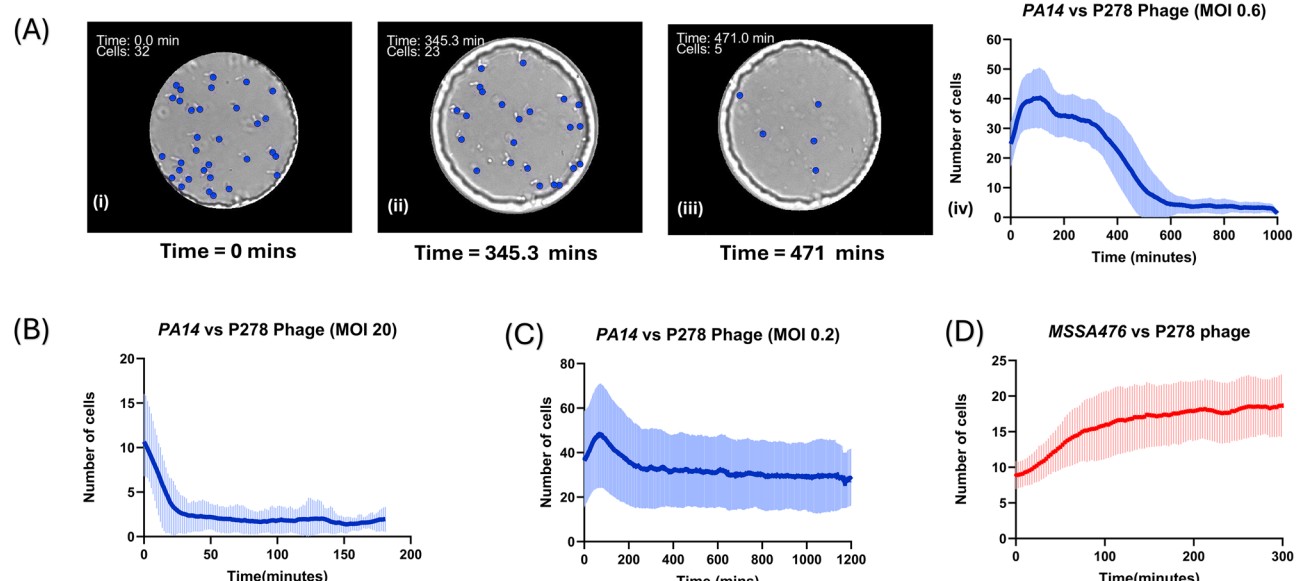

**Fig. 5 | Effect of P278 phage on PA14 ΔflgK and MSSA476 populations at different MOIs. A** Representative brightfield images for MOI 0.6 show droplets at various time points over 500 min, with *P. aeruginosa* cells marked by blue dots cells. **B** Growth dynamics of PA14 ΔflgK exposed to P278 phage at MOI 20. **C** Growth dynamics of PA14 ΔflgK exposed to P278 phage at MOI 0.2. The line plots show the mean cell count over time (solid blue line), with the shaded area indicating the standard deviation. **D** Growth of *S. aureus* MSSA476 in the presence of P278 phage.

The red line represents the mean cell counts over time, with the shaded area showing standard deviation. Supplementary Movie S3 shows the lysis of PA14 ΔflgK cells in droplets at MOI 0.6. Experiments shown in Table 2 were pooled with a total number of droplets screened of 15 for PA14 ΔflgK (MOI 0.6), 6 for PA14 ΔflgK, MOI 20, 6 for PA14 ΔflgK, MOI 0.2 and 10 for MSSA476 +phage. Individual time plots are shown in SI section 7, Supplementary Fig. 10. All error bars represent one standard deviation from the mean.

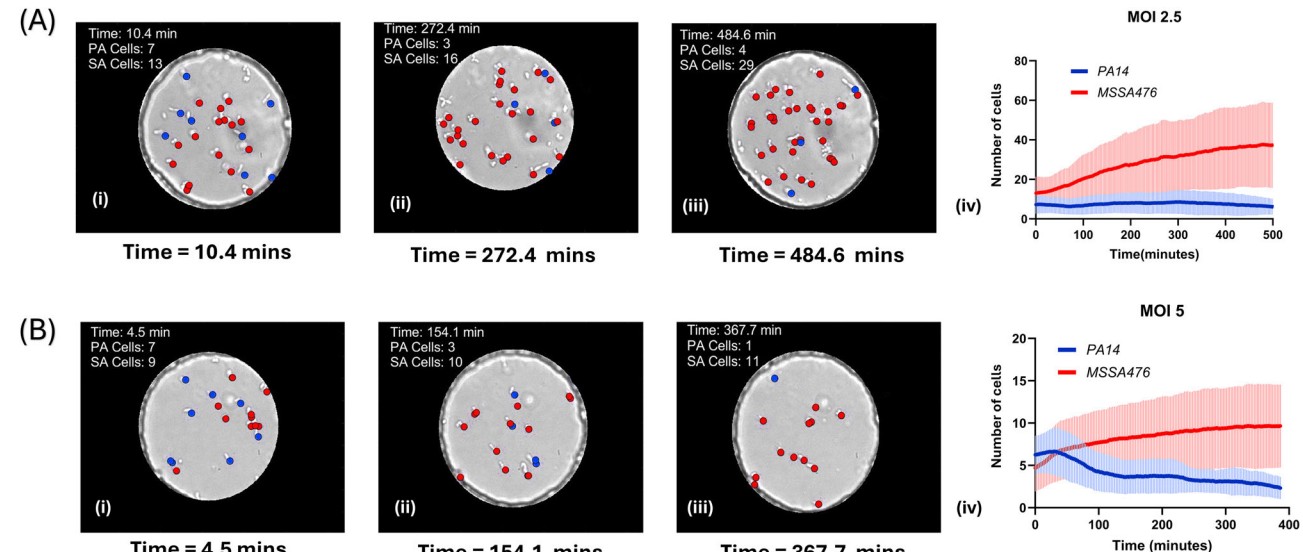

**Fig. 6 | Effect of P278 phage on polymicrobial populations of PA14 ΔflgK and MSSA476 at MOIs 2.5 and 5. A** Polymicrobial growth dynamics of *P. aeruginosa* PA14 ΔflgK (blue) and *S. aureus* MSSA476 (red) in the presence of P278 phage at MOI 2.5 (as shown in Supplementary Movie S4). **A**i–iii Brightfield images at different timepoints highlight the reduction in *P. aeruginosa* cells over time, with S. aureus continuing to grow. **A**-iv Line plot shows the mean cell count over time for both species, with shaded areas representing standard deviations. **B** Polymicrobial

growth dynamics at a higher P278 phage MOI of 5. **B**i–iii Brightfield images at different timepoints highlight the reduction in *P. aeruginosa* cells over time, with *S. aureus* continuing to grow. **B**-iv Line plot shows the mean cell count over time for both species, with shaded areas representing standard deviations. Polymicrobial experiments shown in Table 2 were pooled with a total number of droplets screened of 9 for MOI 2.5 and 9 for MOI 5. Individual curves are shown in Supplementary Figs. 12 and 13. All error bars represent one standard deviation from the mean.

results to ensure fast lysis of PA14 ΔflgK but avoid immediate eradication of the whole PA14 ΔflgK population. Figure 6A and 6B represent the interaction between PA14 ΔflgK and MSSA476 strains in the presence of pseudomonas phage P278 at the two MOIs. The red and blue lines represent the average response of MSSA476 and PA14 ΔflgK strains respectively, with the red and blue shaded region showing the corresponding standard deviations. Individual plots are shown in SI section 9,

Supplementary Figs. 12 and 13. In the first experiment with MOI 2.5, 9 droplets were imaged for over 8 h. The presence of phage P278 was found to suppress the growth of PA14 ΔflgK cells long-term with cell numbers staying at low levels (<5) over the 8 h duration of the experiment. The fastest cell doubling rate for MSSA476 was calculated as 60.21 min with an average cell doubling rate of 100.69 ± 30.08 min. At MOI 5,9 droplets were monitored and the fastest cell doubling rate for MSSA476 was

**Table 3 | Experimental conditions tested and deep learning models used for image analysis for mono and co-cultures of PA14 Δ*flgK* /MSSA476 and exposure to P278 phage with fastest and average cell-doubling rates**

| Experiments | Species analyzed | Culture type | Fastest cell doubling time (minutes) | Average cell doubling time (minutes) | Detection model used |
|---|---|---|---|---|---|
| PA14 Δ*flgK* | PA14 Δ*flgK* | Monoculture | 26.52 | 33.96 ± 6.69 | PA14-v5 |
| PA14 Δ*flgK* + P278 | PA14 Δ*flgK* | Monoculture | – | – | PA14-v5 |
| MSSA476 | MSSA | Monoculture | 41.48 | 69.49 ± 17.78 | MSSA-v5 |
| MSSA476 + P278 | MSSA | Monoculture | 44.39 | 69.16 ± 16.03 | MSSA-v5 |
| PA14 Δ*flgK* with MSSA | PA14 Δ*flgK* | Co-culture | 30.96 | 35.055 ± 4.86 | Bimicrobial-v5 |
| MSSA with PA14 Δ*flgK* | MSSA | Co-culture | 43.39 | 73.34 ± 21.07 | Bimicrobial-v5 |
| MSSA + PA14 Δ*flgK* + P278 (MOI 2.5) | MSSA | Co-culture | 60.21 | 100.69 ± 30.08 | Bimicrobial-v5 |
| MSSA + PA14 Δ*flgK* + P278 (MOI 5) | MSSA | Co-culture | 80.74 | 123.28 ± 37.113 | Bimicrobial-v5 |

calculated at 80.74 min with an average cell doubling rate of 123.28 ± 37.113 min. All extracted cell doubling times across all experiments are summarized in Table 3.

## Discussion

We have established a method for the quantitative analysis of bacteria-phage interactions within model bi-microbial communities in a label-free format. This work relies on the confinement of bacterial communities within stable microdroplets and the ability of deep learning models to learn specific cell morphologies. Further, automation of the imaging platform enabled acquisition of rich imaging datasets yielding over 15,000–20,000 images per experiment. Specifically, this was achieved thanks to the implementation of a deep learning-driven object detection technique to achieve reproducible imaging over long durations tested up to 24 h.

Our method of combining anchored microfluidic droplets with object detection-based image analysis provides several advantages compared to traditional microbiology assays. Unlike absorbance-based liquid (batch) co-culture experiments, which cannot distinguish between different bacterial species, our label-free imaging approach enables accurate differentiation and counting by exploiting morphological differences between species. This eliminates the need for repeatedly plating cells on selective agar plates, or performing other secondary assays, greatly reducing time and resources needed to perform bi-microbial studies.

It is important to maintain focus throughout the duration of experiments to ensure the Z-stack images are taken across the whole depth of the droplets. In this study, we have utilized deep learning-based image classification with 5 classes corresponding to 5 different focal ranges to counter the effects of systematic device tilt and unpredictable focal changes due to constant movement of the automated stage as well as other external factors including table vibrations. The integration of YOLOv8 for focus correction significantly improved image acquisition efficiency and quality. The model achieved a 96% classification accuracy using the training dataset facilitating robust imaging conditions as well as precise focal adjustments. We conducted experiments screening or droplets for over ~20 h with successful integration of this autofocus functionality (Supplementary Fig. 14).

We chose the PA14 Δ*flgK* mutant for its reduced motility, which enabled accurate counting of cells. Highly motile bacteria such as wild-type PA14 (average swimming motility in the range of 45 ± 10 microns per second[37]) would result in multiple detections of the same cell across Z-stacks. With our current time acquisition being 1.5 seconds for 10 slices, we estimate the maximum linear swimming speed of a chosen bacteria for accurate counting to be ~15 μm/s based on the current exclusion sphere method. This also limits the maximum number of cells that can be accurately counted to ~80 cells per droplet. However, such density corresponds to the stationary phase of bacterial growth.

For bacterial detection, we focused on morphology-based analysis as maintenance of cell morphology (especially shape and size) is closely linked

to their fitness[38]. In particular, cells lysed by phages undergo dramatic morphological changes (e.g. explosive lysis, membrane blebbing)[39]. In this study, we observed spheroplast-like round PA14 Δ*flgK* cells formed during lysis. To demonstrate further use of the morphology-based detection method, we have studied the transition of PA14 Δ*flgK* from rod-shaped to such round cell morphology during the phage lysis process (SI, section 11, Supplementary Fig. 15). In most experiments, the number of spheroplasts-like cells in a droplet approximately matched the number of cells being lysed, indicating that the round morphology is a reproducible intermediate shape during the overall lysis process. Significantly, phage particles may remain encapsulated within these spheroplast-like cells, delaying the release of newly synthesized phages and therefore delaying the lysis of other cells. This observation may help in deciphering complex lysis dynamics. Overall, this morphology-based detection method of swimming cells remains currently limited to species/phenotypes with marked differences in morphology.

In this study, we have employed YOLOv5x (extra-large) as it offers several advantages over previous versions of YOLO: less training time (~4x faster training), better mAP precision, ease of implementation, reduced model complexity and transfer learning. Three different models trained on PA14 Δ*flgK*, MSSA476, and bi-microbial datasets achieved detection accuracies of ~99%, ~91%, and ~97%, respectively, facilitating accurate identification bacterial morphologies across different experimental conditions. There remain errors in false positive detections such as dust and traps imperfections and false negative detections for which cells are missed. This occurred frequently when cells were located close to the trap and droplet boundary. In bi-microbial communities, every cell was detected multiple times, and we used the average class to minimise misclassification. In our study, errors in absolute cell counts were reduced through computing a moving average. We also excluded detections which did not move across time series, as they most likely corresponded to false detections.

For PA14 Δ*flgK* monocultures, the average doubling time aligns with literature values in liquid culture assays[40]. MSSA476 alone exhibited slower growth compared to PA14 Δ*flgK* (average doubling time of 69.49 ± 17.78 min). This time was longer than reported values of e.g. 20 min when *S. aureus* was cultured in rich medium[41]. These findings highlight intrinsic growth rate differences in the droplet format, likely influenced by metabolic disparities and environmental adaptation. The absence of mechanical mixing in droplet confinement may also contribute to a reduced effective growth rate of MSSA476. In addition, as visible in Fig. 4B, in absence of agitation, MSSA476 cells tend to remain attached to their parent cells, leading to underestimation of the actual cell number.

In co-culture droplets, however, PA14 Δ*flgK* consistently outcompeted MSSA476, demonstrating a faster average doubling rate compared to MSSA476 and final mean yield ratio of 2.4 ± 1.6 PA:SA. This suggests a competitive advantage for PA14 Δ*flgK*, that has been well reported in literature[10]. However, MSSA476 sometimes outcompeted PA14 Δ*flgK* (i.e. in approximately 10% of our experiments, Supplementary Fig. 5), indicating

that the droplet format results in distinctive growth trajectories for individual species within a polymicrobial setup compared to traditional formats. In our control experiments, PA was found with higher densities than SA in a 24 hour polymicrobial batch culture assay. This may be due to higher oxygen availability as the oil phase dissolves over 10 times more oxygen than water[12,28]. Similar cell doubling rates were observed for PA14 ΔflgK cells in monoculture as well as in a co-culture with MSSA476 cells. However, cell doubling rates for MSSA476 in a co-culture were slightly higher compared to when MSSA cells grew in a monoculture.

Phage P278 exhibited strain-specific lytic activity within droplets. PA14 ΔflgK's growth was significantly suppressed across all MOI conditions, with rapid lysis observed at MOI of 20 indicating that higher MOI leads to faster lysis of cells. The initial increase in cell number in experiments with MOI 0.6 and 0.2 can be attributed to the latent period being longer than cell division rate for PA14 ΔflgK in a monoculture. The fastest cell doubling rate for PA14 ΔflgK cells in a monoculture (~27 min) was found to be lower than the time to lysis as observed in the one step growth curve (~37 min). In contrast, MSSA476 was not affected by P278, maintaining growth rates comparable to phage-free conditions (~70 min in both cases). In co-culture experiments, phage P278 selectively inhibited PA14 ΔflgK, allowing MSSA476 to outcompete PA14 ΔflgK. At MOI 2.5 and MOI 5, MSSA476 displayed long doubling times of $100.69 \pm 30.08$ min and $123.28 \pm 37.113$ min, respectively. We hypothesize that these extended cell doubling rates could be due to the presence of PA14 ΔflgK cell lysate and high numbers of phage particles within each droplet. Another hypothesis is that phage synthesis by infected PA14 ΔflgK cells may lead to higher intake of nutrients from the outside environment, depleting them for MSSA476 cells[42]. Additional co-culture experiments at different MOIs could disentangle the influence of initial phage and PA14 ΔflgK numbers on MSSA476 growth profile.

The droplet format helps uncover the heterogeneity of growth and responses to phage exposure. Thanks to stochastic cell loading, we generated droplets with a large diversity in initial cell numbers per species. The diversity of growth curves is illustrated in the individual plots shown in SI and can help shine light on assay reproducibility and the statistical significance of results. This enables the study the inoculum effects, i.e. the influence of initial bacteria densities on the efficacy of phage treatment. For example, the final yield for PA14 ΔflgK with P278 at MOI 0.2 was positively corelated with the initial cell number. Likewise, phage MOIs represent an average number for phage dosing, and absolute numbers of phages in each droplet will vary. To date, inoculum effects in droplets have only been studied in the context of response to antibiotics[43].

To conclude, the deep learning bacterial detection framework enabled us to differentiate PA14 ΔflgK and MSSA476 in a mixed confined 3D environment based on morphological features. This capability is critical for understanding species-specific dynamics and interactions. The ability to observe and quantify bacterial dynamics in real time provides invaluable insights into growth patterns, competition, and antimicrobial efficacy. This study used morphologically distinct strains (rods vs cocci) as a proof of concept. For closely related or similarly shaped species, morphology alone may not suffice, especially given cells have varied orientations in liquid phase. Misclassification rates would likely rise with >2 species unless paired with additional labelling (e.g. fluorescence) or phenotypic signatures. In future work, expanding to instance segmentation, or hybrid fluorescent-morphological approaches could mitigate these challenges. To extend the current study to more motile strains and higher cell numbers, one has to increase the speed of the current Z-stacking implementation, avoiding slowly moving mechanical parts, e.g. making use of piezoelectric Z-stages, piezo-objective scanners or electrotuneable lenses.

The presented methodology can be extended to study other microbial and phage systems, enabling a deeper understanding of pathogen dynamics and co-evolution mechanisms. Additionally, it provides a robust platform for testing new antimicrobials and phage therapies in in-vitro polymicrobial settings. Our approach will be also useful to study the dynamics of bacterial communities in response to external physical, chemical and biological

stressors. In future, other phenotypes could be studied in addition to cell localization such as morphological adaptation in response to stress or change in motility patterns[44,45].

By overcoming the drawbacks of traditional microbiological polyculture assay methods, requiring repeated plating experiments or lengthy molecular assays over long timescales, our strategy offers rapid quantification of multiple cell species label-free. We are able to track species-specific dynamics, detect cell lysis events, and analyze polymicrobial-phage interactions at single-cell resolution. Even though the present study is small scale in terms of droplet numbers, the method can produce rich imaging datasets in excess of 20,000 images per droplet, yielding over 200,000 images per experiment, which are processed into millions of detections across multiple timepoints.

Future work could include the study of larger multi-species communities. This could be done by integrating additional imaging modalities (e.g., using fluorescent strains) and expanding deep learning models to include other bacterial species and morphological variants. The findings highlight the potential of phages as targeted antimicrobial agents, particularly in polymicrobial infections where long-term *Pseudomonas* growth suppression is critical to outcomes in patients with chronic infections.

## Methods
### Bacterial strain and phage lysate preparation
*Pseudomonas aeruginosa* strain PA14 ΔflgK (gram negative) and *Staphylococcus aureus* strain MSSA-476 (gram positive) were chosen as the model strains for this polymicrobial study. Lysogeny Broth (LB) agar plates with single colonies of each strain were obtained from glycerol stocks stored at -80° C. To initiate an experiment, a single colony from each plate was picked and added to a sterile culture tube containing 5 mL of LB media (10 g/L tryptone, 5 g/L yeast extract, 10 g/L NaCl) and incubated overnight at 37° C with shaking at 200 rpm for at least 14 h without phages. Overnight culture was then added to 5 mL of fresh LB media for further cultivation until a desired $OD_{600}$ (optical density measured at 600 nm) was obtained depending on specific experimental requirements.

The number of cells in droplets follows Poisson distribution. This allows us to predict the average number of cells in each droplet system based on $OD_{600}$ values, which relates to colony forming units per mL. We obtained an $OD_{600}$ versus CFU/mL calibration curve for both species to estimate the average number of cells for a given $OD_{600}$ value (SI, section 12, Supplementary Tables 3 and 4, Supplementary Figs. 16 and 17). A broad range of experiments were conducted to obtain data on individual strains as well as their combination starting at different $OD_{600}$ values to study strain fitness. Desired $OD_{600}$ values for growth experiments were chosen between 0.100-0.400 to encapsulate fewer than 5 cells in each droplet. Lysis experiments were conducted at $OD_{600}$ values between 0.6-0.8 to ensure that each droplet had at least 10 cells. Table 1 shows all experimental conditions for growth experiments. Standard colony forming unit (CFU) assays were also carried out to obtain specific population numbers pertaining to the $OD_{600}$ values.

Phage lysate P278 was preserved in SM (Storage and Maintenance) buffer (0.1 M NaCl, 8 mM MgSO4·7H2O, 50 mM Tris-Cl, pH 7.5, 0.01% gelatin) at 4° C, with phage titer of $8 \times 10^9$ PFU/mL (PFU, plaque-forming unit). For single strain as well as polymicrobial lysis experiments, different volumes of strains cultured in exponential phase were mixed with the cell lysate to obtain desirable multiplicity of infection (MOI). MOI is defined as the ratio between the number of phage particles and number of bacterial cells. Table 2 shows all experimental conditions for phage-induced lysis experiments.

### Microfluidic chip fabrication
The two-layer microfluidic device was produced using soft lithography and a high-resolution acetate mask (Microlithography Services). The procedure involved patterning negative photoresist SU8 TF6000 (MicroChem) on a silicon wafer, which was exposed to UV light through a transparent film mask. The device comprises two layers: the first layer includes 3 μm-high pillars, while the second layer contains circular microdroplet traps, also 3 μm

in height, with a diameter of 60 μm. Further details on the procedure can be found in a previous publication[29]. A mixture of PDMS (poly-dimethylsiloxane, Ellsworth) and a curing agent at a 10:1 ratio was poured onto the patterned wafer after being degassed. To prevent droplet evaporation during time-lapse experiments, a small coverslip (0.15 mm thick) was placed over the trap array before PDMS curing. The wafer was cured at 70° C for 120 min. After curing, the PDMS was cut out and 1 mm inlet holes were punched (Kai Medical) for oil and bacterial culture media. Plasma treatment (Diener Zepto) was used to bond the PDMS to a thin coverslip (22 × 50 mm, 0.13–0.17 mm thick). Finally, a 1% (v/v) solution of tri-chloro(1H,1H,2H,2H-perfluorooctyl) silane (Merck) in HFE-7500 oil was flushed through the device, followed by incubation at 70° C for a minimum of 30 min.

### Anchored droplet generation and trapping
The microfluidic device was positioned on the microscope stage and secured with scotch tape. Droplets were formed on-chip through phase change, where the aqueous phase containing bacterial cells or a bacteria-phage mixture was replaced by the oil phase, creating anchored droplets at the designed traps. The continuous phase was composed of 1% (w/v) 008-Fluorosurfactant (RAN Biotechnologies) in HFE-7500 oil (Fluorochem). The aqueous phase included either bacterial strains in LB broth for growth experiments or bacterial strains in LB broth with P278 phages for lysis-based experiments. A schematic of the microscope setup is shown in Fig. 1.

To initiate the experiments, the aqueous and oil phases were loaded into PTFE tubing (SLS) connected to 1 mL plastic syringes (BD Plastipak), which were mounted on syringe pumps (Nemesys, Cetoni). A volume between 100–200 μL of bacterial culture or bacteria-phage mix was aspirated into one syringe for injection into the chip. The device was first primed with oil to expel any air from the trapping chambers. Once the oil flow was stopped, the bacterial or bacteria-phage solution was introduced until the entire trapping array was filled (Fig. 1B). The flow of the bacterial solution was then halted, and oil was reintroduced to flush the cell sample, creating droplets of the immobilized cell sample in the circular traps.

### Imaging and focus correction using deep learning
An inverted Olympus IX73 microscope with a motorized XY stage was used to image the bacterial cells trapped in different chambers on chip. The microscope was equipped with a white LED (CoolLED pE-100) and featured a bolt-on motorized Z-focus drive (PS3H122R, Prior Scientific) controlled by a ProScan III focus controller (Prior Scientific). Imaging was conducted using a 40x objective with a numerical aperture of 0.45 (UPLFLN40X-2, Olympus). The microscope was installed on a vibration-damping platform (Newport VIP320X1218-50140). A monochrome industrial USB camera (DMK 37AUX287, The Imaging Source) was used to capture images of the droplets at a resolution of 640 ×480 pixels, with 8 bits encoding. Exposure time for transmitted light imaging was 19 μs. X and Y coordinates of selected droplets to be imaged within the droplet array were stored in a text file using a python script. Using another custom python script, the camera captured Z-slices spanning from above to below the best focal plane to generate a 3D stack for each droplet. The motorized stage and motorized Z-focus were used for precise X, Y, and Z movements during the imaging process with customizable focal adjustment steps.

To ensure consistent Z-stacks across all droplets, a pre-trained YOLOv8 deep learning-based image classification model was used to analyze initial droplet images and determine the current focal level. The model was specifically trained to classify focus levels into five categories (Above-Greater3, AboveLess3, InFocus, BelowGreater3, BelowLess3) as explained in Fig. 2.

The use of two different classes for slightly out-of-focus (<3 μm) and greatly out-of-focus (>3 μm) both above and below the plane allowed for greater accuracy in detecting location in the Z-stack, as droplets slightly out of focus in either direction were visually more similar to each other than very out-of-focus drops, regardless of direction. The categories were then remapped to an integer between 1 and 5 based on their respective classes for

focus adjustment purposes. The system then computed a score weighted by confidence values based on model outputs to determine how far the image was from the best focus. We used the following equation to compute the score:

$$S = \sum_{c=1}^{5} c.w_c \tag{1}$$

Where $w_c$ is the confidence for each class c. If the score belonged to a predefined range of $2.8 \le S \le 3.5$, the image was considered 'in-focus', as found empirically. Otherwise, the stage was adjusted by a pre-defined step size (typically 0.8 micron), and a newly acquired image was used to calculate an updated S score until optimal focus was achieved. The objective was moved upwards by 0.8 μm for scores below 2.8 and 0.8 μm upwards for values above 3.5.

The system recorded a history of focus movements over time (SI section 10, Supplementary Fig. 14). If the focus drifted beyond a pre-defined maximum number of iterations (corresponding to approx. 4 microns) for an extended period, the system detected this as focus instability and reversed the focus direction to attempt refocusing. Once focus was attained, a Hough transform was employed to detect the circular droplet trap and correct any axial drift of the droplet's center position. The system first captured an image of the droplet and applied the Hough transform to identify circular patterns within the image. The detected center of the trap was compared to the expected initial position, and the system computed the necessary x and y adjustments to align the micro-scope stage. The stage was then moved to these updated coordinates, ensuring accurate droplet positioning over time (see Supplementary Movie S1). The Python script also allowed modification of number of time points, the duration between each time point as well as number of images and spacing between slices in the Z-stack for each time point. This method enabled robust, automated droplet imaging with real-time corrections for focus drift and axial misalignment, ensuring high-quality imaging data over long experimental durations of over 12 h.

### Deep learning model generation for cell morphology detection
YOLOv5 (You Only Look Once, Version 5) is a single stage object detection model based on convolutional neural networks (CNN) developed by Ultralytics[46]. It is widely used for detecting objects of interest in an image or a video at high-speed and accuracy. YOLOv5 was chosen for object detection-based applications for this study as it outperforms many versions of object detection models such as YOLOv4, Mask R-CNN, R-CNN, RetinaNET and Single Shot MultiBox Detector[47].

YOLOv5x (extra-large) was chosen as the base model as it provides the highest detection accuracy compared to other models in the YOLOv5 family. In this study, we trained 3 different models of YOLOv5x for detecting bacterial strains with different morphologies (see Table 4). Some of the models were trained using transfer learning. Transfer learning is a machine learning technique where a model developed for one task is reused as the starting point for a model on a different, but related, task. The models were trained on the Google COLAB webserver and the training was completed within a few minutes. All training parameters are reported in the supplementary section 2 with mAP, training and validation specifications.

The first model was trained for detecting the PA14 Δ*flgK* strain with rod shaped morphology (Fig. 3A-i). The dataset was generated by taking images from PA14 Δ*flgK* growth and lysis experiments. A total of 230 images containing over 1600 examples of PA14 Δ*flgK* morphology in different microscopic conditions (light intensity, gain, collimator position) were labelled manually by drawing bounding boxes. This dataset was then split into a ratio of 70:30 for training and validation respectively. The second model was trained detection of MSSA476 with cocci morphology or forming figures of 8 or aggregates with figures of 8 (Fig. 3A-ii). A total of 150 images containing over 1300 examples of MSSA476 morphology in different microscopic conditions were labelled and split into a 70:30 ratio like the PA14 Δ*flgK* model. The third model was trained for detection of both

**Table 4 | Training parameters for each model**

| Model | No. of training images | No. of PA cells | No. of MSSA476 cells | Epochs (training 1) | Epochs (training 2) |
|---|---|---|---|---|---|
| PA14 Δ*flgK* | 230 | 1600 | - | 200 | - |
| MSSA476 | 160 | - | 1300 | 200 | 50 |
| Bi-microbial | 480 | 2000 | 2200 | 150 | 80 |

PA14 Δ*flgK* and MSSA476 cell strains. The dataset consisted of 480 images with over 2000 examples of PA14 Δ*flgK* morphology and 2200 examples of MSSA476 morphology. This dataset was generated by combining the first two datasets as well as adding examples from polymicrobial experiments where both PA14 Δ*flgK* and MSSA476 cells were present in one droplet.

## Two-species bacteria detection using YOLOv5

We developed a counting method to extract simultaneously the number of cells for PA and SA per droplet using Z-stack imaging. The Z-stack images collected using the imaging setup were utilized to perform morphology-based detections. Detection data from each image contained information on bounding box coordinates, detection scores and class information which were stored in corresponding detection files (Fig. 3B). All image processing was performed using MATLAB. For each time point, the corresponding images were loaded from a local folder, while the bounding box detections and class information were imported from detection files. Each Z-slice within a time point was analyzed for cell counts and classifications (Fig. 3C). Bounding box coordinates were calculated based on image dimensions (640×480 pixels). Cells were classified into PA or SA based on the class label and detection score. Only detections with confidence scores above a threshold of 40% were considered valid to eliminate low-confidence detections.

To account for bacterial movement between slices, we calculated the Euclidean distance between bacteria across different slices. If two bacteria belonging to different Z-slices were detected within a predefined distance threshold (typically within a sphere of diameter 3 microns, which was determined using movement speed of the bacteria), one detection was excluded to avoid counting the same bacterium multiple times. In the case of 2 classes of cells, namely PA and SA, cells were detected accurately thanks to their marked difference in morphology. Occasionally, one cell was classified differently in two slices within the same stack, in which case the cell was classified based on the overall proportion of PA versus SA detections. Cells with a ratio over 50% of PA detections were labeled as PA, and vice versa. This classification method allowed real-time counting and localizing of PA and SA populations within droplets (Fig. 3). Time assignment was performed by extracting the metadata from the first image in the sequence, which provided the initial time point. For subsequent images, we calculated the elapsed time in seconds relative to this initial time point using file metadata.

## Cell doubling time calculation

Doubling times for bacterial populations were calculated by analyzing the slope of the bacterial growth curves. To ensure accuracy, we identified the portions of the plots with the steepest slopes within a 20 min moving window. Some examples of such moving windows and corresponding maximum slopes are provided in SI section 13, Supplementary Figs. 18 and 19. A MATLAB script was utilized to calculate the maximum specific growth rates (SR) based on the maximum slopes of the selected regions. These specific growth rates (SR) were then used to derive the cell doubling times ($T_D$) using the formula:

$$T_D = \frac{\ln(2)}{SR} \tag{2}$$

## Statistics and reproducibility

The number of technical and biological replicates for all experiments are mentioned in figure legends. Numerical data shown in Figs. 4, 5 and 6 represent the mean values of these replicates ± standard deviations. Data was analyzed using GraphPad Prism 10 (GraphPad Software, CA, USA).

## Data availability

All numerical data for all the plots in this study are available in the following repository: https://github.com/GielenLab/YOLOv5-v8-polymicrobial. Training images and model weights data will be shared upon reasonable request.

## Code availability

Code used for YOLOv5 and YOLOv8 training are available on https://github.com/ultralytics. All the MATLAB scripts used for analysis, python code for Z-stack acquisition and drift correction are available in the Github repository: https://github.com/GielenLab/YOLOv5-v8-polymicrobial.

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

## Acknowledgements

The authors thank Dr. Nela Nikolic, Dr. Tobias Bergmiller, Dr. Vasileios Anagnostidis for useful discussions and comments, Prof. Ben Temperton for access to the Exeter Citizen Phage Library, Prof. Stefano Pagliara for the bacterial strains. We acknowledge the LSI Technical Services Team at the University of Exeter and use of the Exeter Microfluidics Facility and Savchenko Centre for Nanoscience. For the purpose of open access, the author has applied a 'Creative Commons Attribution (CC BY) licence to any Author Accepted Manuscript version arising from this submission. This work was supported by the BBSRC grant BB/T011777/1 to FG, a NERC Cross-disciplinary Research for Discovery Science grant to RM and FG. We acknowledge support from the EPSRC VIS to AMD. This work was also supported by the Biotechnology and Biological Sciences Research Council-funded South-West Biosciences Doctoral Training Partnership [training grant reference 2578821].

## Author contributions

A.T., F.G. and R.M. conceived and designed the study. A.T. performed the experiments. A.T. and A.M.D. contributed to data analysis and software conception. A.T., A.M.D., R.C., R.M. and F.G. helped with project design and discussions. A.T. and F.G. wrote the manuscript. A.M.D., R.C. and R.M. reviewed the manuscript. All authors read and approved the final manuscript.

## Competing interests

The authors declare no competing interests.
