## [Transparent Peer Review file · Communications Biology]

Harnessing droplet microfluidics and morphology-based deep learning for the study of polymicrobial-phage interactions

Corresponding Author: Dr Fabrice Gielen

Version 0:

Reviewer comments:

Reviewer #1

(Remarks to the Author)

This paper describes a method to analyse the growth of bacteria cultures using droplet microfluidics and an AI-based autofocus function together with morphology-based deep learning models. The experimental system itself has already been published before by the authors (doi:10.3389/fnlct.2023.1258155), however, they extended that with a nice and useful method to study and analyse the growth of polymicrobial communities. The techniques interesting, and could be of interest for the microbiologist community. A limitation of the method is that only the interaction of bacterial strains possessing different morphologies can be studied.

The method itself is promising and could be of interest in the research field, however I have some concerns addressed to the authors. Those should be clarified to see what are the limitations and how robust the system is. Also, they are needed in order to be able to reproduce the experiments.

Major comments:

- My major concern about the manuscript is that the CFU and MOI values are not consistent. In the first part of the paper the authors describe the growth dynamics of two different bacterial species within the droplets. The OD values are summerized in Table 1. However, the authors should also give the CFU values. They use two different species, having different morphologies. The same OD value for *P. aeruginosa* might result in different cell number compared to *S. aureus*. This actually could also be seen in the example presented in Figure 4, both on the images and the graphs presenting the average cell numbers. I wonder if this could be the reason for the longer lag phase of PA14 compared to MSSA476.

In Table 2. The authors summerize the conditions of the lysis experiments. Here The MOI values are not clear. When they perform lysis experiments with PA14 strain, they claim to use MOI 20, 0.6 and 0.2. However, I can see where MOI=20 comes from, but based on the cell numbers they give, the MOI value should be around 13 and 15 for the other 2 cases. So how did the authors calculated those values? It is a crucial part of the paper. Also, When they studied the interaction of SA with the phage, they say MOI=0.2 was applied, however, they do not give the CFU numberf for that strain. I have the same problem with the polymicrobial experiments. CFU-s are not given for the SA strain, and the calculated MOI values are not consistent with the ones above. This part should be clarified within the paper.

- The authors present only the first couple of hours of the experiments, however they say that they tracked the growth of the culture for about 20 hours. To get proper growth curves, it would be nice to see how the cell number changes in long term. Actually, this would give evidence on how many cells could be analyzed with this method.

- It would be important to clarify how many individual expeirments were performed by the authors for each scnearios. Sometimes it is written how many droplets were analyzed, but not all the time. Authors should write how many droplets and how many biological replicates were performed. Also, n should be indicated on the graphs or in the captions.

- In section 1.7 "Cell doubling timw calculation:

In this section authors should write down in more detailes how they calculated the doubling time. It is not clear how the slope was calculated, which part of the curves were used, also how the authors define growth rate.

- How did the authors calcualted the burst size? They say that the burst size was calcualted to be 46, but there is no

evidence how this was determined.

- The used mutant strain has a reduced motility, this can also be seen in the videos. It shows mostly brownian motion. What does happen if we would like to study a swimming strain? I have the feeling that this method can not be used in such cases. Also, what happens when the culture grows? Is there a maximum number of cells above which the tracking method does not work? How large (how many cells) is the biggest cell culture that can be analyzed with this method? Authors should add the limitations of this method into the discussion of the manuscript. Adding the growthcurves for prolonged periods, not just for 3-5 hours, would also help in seeing the efficiency of this method.

Minor comments:

- There are figures and tables that are not referred in the text (e.g., Table 2, Table5, Figure 2). Also, figures and tables should be positioned after they are first referred in the text. Author should go through the paper and check that all figures and tables should be referred. This is also true for the content of the supplementary informations.

- The figure numbers in the supplementary informations are shifted. Figure S9 is referred as Figure S8 in section 2.1.6, and figure S7 is referred as Figure S6 in section 2.2.

- The authors should use more references. There are claims in the introduction that are not supported by any references. E.g., "For example, rapid resistance acquisition and high fitness of resistant bacteria have often been reported when using monocultures." The authors only have 36 references, so they could add some more to support such statements.

- In the methods section author should define "overnight culture". How many hours did they culture the cells for the overnight cultures? It could be important when setting the initial cell number. Since they talk about the interaction of bacteria and bacteriophage, it is important to reduce the possible number of resistant bacteria in the initial culture, therefore the duration of the overnight culture and its dilution could be important in these experiments.

- In section 1.4, first paragraph: authors use the term dynamic environment, however the environment that the cells encounter with can not be changed throughout the experiment. Therefore, this term is misleading.

- In section 1.5, fourth paragraph: authors refer to "Figure3-ii", but I think they should use Figure 3A-ii.

- In section 2.1.2, first paragraph: Authors write "The flgK mutants had low motility..." In this sentence they should write flgK in italic or with a capital F (FlgK).

- In section 2.1.2: There is no need of Table 4, those results can be summarized in one sentence.

- I do not understand the Figure S5 in SI file. There is no description of it.

Reviewer #2

(Remarks to the Author)

Reviewer report

Communications Biology

"Harnessing droplet microfluidics and morphology-based deep learning for the study of polymicrobial-phage interactions"

In this manuscript, the Authors present a method of image analysis based on machine learning algorithms where the algorithms are trained on morphologies of two bacterial species, one rod and one coccus. In addition, the Authors present an algorithm that autofocuses the acquired images. The Authors show how they employ their methods to image phage infections of bacteria in droplets.

This is a proof-of-principle technology development paper with added validation of the methods presented and does not show any new interesting results in terms of phage-bacteria interactions. I think that the presented image analysis pipeline might be useful for droplet-based assays but is quite not there yet. The Authors use droplets, but they only use 20 droplets in an assay – it does not make sense to involve droplets at this scale. The Authors mention that the "traditional" assays require a single layer of cells to work, but their droplets are essentially flat (6 μm high). The autofocus algorithm that the Authors present seems to be doubling the function of existing hardware-based microscope modules (such as Nikon's Perfect Focus or Leica's Adaptive Focus Control) or even software-based autofocus modules, thus I am unsure of its novelty – albeit it might be useful in low-resource settings.

Some specific comments:

- 1) A major issue: Why did the Authors use ΔflgK mutant for *P. aeruginosa*? They say it is because it has impaired mobility in comparison to wild type and is thus easier to run image analysis on it. What are the limitations of the presented image analysis algorithms in terms of motility of analysed bacteria? The Authors account for bacteria movement between slices in chapter 1.6. Have the Authors prepared an analysis of bacterial movements at a high frame rate of imaging to establish that 3 μm diameter sphere is a good basis to assume that a bacterium will not move more than this distance between frames? Does it mean that the highest number of cells analysed in a droplet is determined by the number of virtual spheres around cells, each sphere 3 μm in diameter?

- 2) How would the system perform if a droplet was not flat but spherical of the same volume as the Authors used, 11 pl?
- 3) The Authors compare their system to traditional methods of analysis of polymicrobial cultures. Can the Authors prepare a similar brief comparison to state-of-the-art microfluidic systems so that the reader has a better oversight on the novelty of the presented method? There are labs of Oscar Hallatchek, Jeff Hasty, Howard Stone, Bartlomiej Waclaw, late Alma dal Co, Simon van Vliet...
- 4) The Authors say that "most" current imaging methods require cells to grow in single layers – what are the current imaging methods that do not have this requirement? I think about the research done in 3D in the group of Knut Drescher (<https://www.nature.com/articles/s41564-020-00817-4>), but there may be others?
- 5) I agree that quantification of motile cells in droplets is more challenging technically than of cells in biofilms or in single layers. The Authors, however, do not make a strong case about why it is important to be able to study cells in liquid phase instead of on surfaces, especially that they do not perform their later described experiments at a large scale in droplets (they use up to 20 droplets), thus not utilizing the possibility of screening a large parameter space which would be a major advantage of droplet use over traditional methods.
- 6) Microbial interactions have, indeed, been studied in droplets, also label free and at a large scale: Kehe et al. <https://www.pnas.org/doi/10.1073/pnas.1900102116>
- 7) I admire the Authors' honesty about using scotch tape to immobilize microscopy samples. I use scotch tape and blu tack.
- 8) Why is detection of PA14 more reliable than of MSSA476 as per Table 4?
- 9) Why there are seemingly PA14 bacterial cells that survive (not a rare event, as in Fig. 5B the mean number of surviving bacteria even including standard deviation is above 0) MOI 20 of P278 as in Fig. 5?
- 10) Why is the number of analysed droplets 6, 10, 20? Why does the method not support high throughput image analysis of thousands of droplets? This raises questions about the claim in discussion that the presented method presents "large datasets for both technical and biological repeats".
- 11) The Authors say that by targeting an average of 10 cells per droplet, some cells contained as little as one bacterium. By using Poisson distribution at expected value of 1 and mean of 10, we can see that 0.45% of droplets should contain one bacterium when the mean was 10. As the Authors use 20 droplets per experimental condition at most, it is unlikely that any of their droplet contained a single bacterium.
- 12) Inoculum effect of antibiotics on bacteria has already been studied in droplets, it is fair to cite as no explanation of inoculum effect by the Authors is introduced. Postek et al. <https://pubs.rsc.org/en/content/articlelanding/2018/lc/c8lc00916c>

Reviewer #3

(Remarks to the Author)

The authors co-cultured *Pseudomonas aeruginosa* and *Staphylococcus aureus* inside a single droplet and monitored their relative growth kinetics for up to 24 hours on an imaging basis with deep learning. They challenged *Pseudomonas aeruginosa* lytic phages with monocultures and multicultures and quantified the growth rates, relative bacterial density, and lytic kinetics of the two species. The content is very interesting and unique. I would like to suggest that this article will be published in *Communications Biology* after major revision. The author should address followed concerns.

(1) As a basic rule, when data or schematic diagrams presented in Figures are described in the text, it should be stated exactly which Figure they correspond to. It will aid the reader's understanding. Please correct the following :

- Which sentence applies to the explanation of Figure 2, starting around L211? Correct it.
- In L255, is Figure 3-i a mistake for Figure 3A-i? Please state exactly. Similarly, in L261, Figure 3-ii should also be stated correctly.
- Which sentence corresponds to Figure 3B and Figure 3C? Correct each.
- Which sentence corresponds to Figure 4C? It probably corresponds to Section 2.1.6. Correct it.
- Which sentence corresponds to Figure 6B? It probably corresponds to Section 2.2.3. Correct it.

(2) I got the impression that the discussion after paragraph starting from L537 is lacking. In particular, the discussion of the series of microbial growth in the droplet is poor. Please respond to the following points.

- L541: These findings highlight intrinsic growth rate differences in the droplet format, likely influenced by metabolic disparities and environmental adaptation.

The authors cite metabolic disparities and environmental adaptation as reasons for the 20-minute slower SA doubling time in droplets than in bulk culture. Of course, since microorganisms are confined to the microscopic space of a droplet, adaptation to the environment and the metabolic changes that result from it would be well taken into account. However, the authors' experiment is the same in that they used a rich medium, which is considered to be an adequate environment for microbial growth. On the other hand, are differences in culture methods considered as a factor? For example, in reference No. 34, they use concussion culture. The concussion culture has the effect of aeration. Regarding aeration, in this experimental system, sufficient oxygen concentration is provided, as the authors describe that the oil phase dissolves 10 times more oxygen than water. Concussion culture also has a role in physically aiding cell division. Droplet cultures are static cultures, which eliminates the physical effects of shaking, but do you consider the possibility that this may have slowed down the cell doubling time?

- L546: However, MSSA476 sometimes outcompeted PA14 (i.e. in approximately 10% of our experiments, Supplementary Figure S9)

Please add which droplet in Figure S9 corresponds to this phenomenon.

- L547-54: the droplet format is more conducive to balanced growth for the two species compared to traditional formats.

Even though SA doubling times are different from bulk cultures, Why are droplet cultures better than bulk cultures? The authors state that this is in comparison to bulk culture, but are they also co-culturing bulk in this study? If the authors are doing experiments, shouldn't they present that data for discussion? Or are they trying to argue that they are better suited to looking at the growth of individual microorganisms as opposed to bulk culture? It would be better to state exactly what you mean, as it could be confusing to the reader.

- L564: We hypothesize that these extended cell doubling rates could be due to the presence of PA14 cell lysate and high numbers of phage particles within each droplet.

Why would a large amount of phage particles increase the cell doubling time? Assuming that, we can examine this by culturing MSSA476 together with a large amount of phage in droplets, as shown in Figure 5d. Why has that experiment not been performed?

Furthermore, in the absence of competitor P14, would MSSA476 not be inhibited in growth, allowing it to grow at the same doubling time under the same conditions as when cultured monoculture? For example, if P14 and MSSA476 are co-cultured under MOI 20 conditions, it is assumed that P14 is immediately lysed and the phage itself cannot be increased. In this case, is the doubling time of MSSA476 normal? Co-culturing P14 and MSSA476 under different conditions, such as MOI 20 or 0.6 in monoculture, would help further discussion on the effect of the number of phage particles present on the doubling time of MSSA476.

Are there any other possibilities? For example, since it takes a lot of energy for phage to increase within P14, many media components are utilized, reducing the media components available for MSSA476 to proliferate. A disadvantage of droplet culture is that the droplet volume is pL scale, so it is quite possible that nutrients will become depleted and the growth will stop. I think it would help researchers who deal with microorganisms if we discuss this point of view as well.

(3) In conclusion, the authors discuss the potential of this technology but do not discuss the problems and limitations of the current situation.

It would be appropriate to add a discussion of the limitations of distinguishing endemic species from microbial morphology by imaging. Since PA14 and MSSA476 are clearly different in morphology, as in this study, this would have been good material for an initial POC to demonstrate the authors' concept. However, with the numerous number of microbial species in the natural environment, is there a limit to morphological classification? For example, is it possible to distinguish between different species of microorganisms that are very similar in morphology? In addition, when it comes to distinguishing between three or four microorganisms, the difficulty of identifying microorganisms instantly increases, and it is easy to imagine how difficult it would be to distinguish between them. The authors should be a bit more careful in describing, as it could mislead the reader.

Version 1:

Reviewer comments:

Reviewer #1

(Remarks to the Author)

The authors addressed all of my concern, the revised manuscript contains detailed information about the experiments and also the discussion of the manuscript became more prudent.

I have some comments that should be addressed, but after these corrections I suggest the MS to be accepted for publication (minor revision).

1) In the abstract the authors claim that their method is suitable to follow the growth dynamics of *P. aeruginosa* and *S. aureus* for 24 hours. I feel that this is misleading, and it is an overstatement. The authors can only follow the killing curves of the phage for long-term. If cells are growing they mention that there is a limitation in cell number to follow... Also there are not too many graphs presented in the paper with the time scale of 20 hours... Most of the growth curves were followed for 4-5 hours. So, the authors should remove this sentence ("We show that we can monitor the relative growth dynamics of *P. aeruginosa* and *S. aureus* growing in 11 picolitre droplets for up to 24 hours.") from the abstract

2) The CFU graph of PA14 mutant strain and the data in Table S1 are not consistent in the Supplementary Informations. Also the values on the y axis...I guess they should be multiplied by 108 or 109 ? And it is also not consistent with the data presented in Table 1 in the manuscript. Please check these data and correct it! For the other bacterial strain (MSSA476) it is ok.

3) I see no reason why the last chapter of the Supplementary Informations (Comparison to microfluidics-based studies) is not included in the MS. It would be nice to add such concept into the discussion and/or introduction. Maybe leaving the table S4 in the supplementary file, but adding the discussion part into the manuscript. Please consider it.

Reviewer #2

(Remarks to the Author)

I think that the Authors have now clearly stated the limitations of their system, and with this narrowed scope the method is honestly described.

I believe that the method itself is a nice proof of concept that might lead to actually high-throughput single-cell level interaction studies of different bacterial species (e.g. more droplets analysed). Until now, cells had to be either immobilised in a hydrogel or on a surface to warrant single-cell level measurement of growth over time. This is sometimes a limitation in studies of bacteria in droplets, and the Authors circumvent that. Again, this is not an actually high-throughput method (even though it generates tons of data), but it is moving in the right direction.

I think this manuscript is suitable for publication in Communications Biology.

Perhaps it would be good if the Authors placed figure R2-1 in the Supplementary Information as it addresses a question of precision of bacterial detection.

I wish the Authors all the best in their future work, and I thank other Reviewers and the Editor for their input.

Reviewer #3

(Remarks to the Author)

The revised manuscript surely has responded to all of the suggestions, comments, and questions that I had sent. Thus, I would like to recommend that the manuscript now would be accepted for publication in Communications Biology.

Dear Reviewers,

We would like to thank you for the time and effort invested in reviewing our manuscript COMMSBIO-25-2729A, titled “*Harnessing droplet microfluidics and morphology-based deep learning for the label-free study of polymicrobial-phage interactions.*”

We greatly appreciate your constructive feedback, which has been instrumental in improving the quality and clarity of our work. In particular, we have strengthened the justification for studying polymicrobial planktonic cells in label-free format. We have added extended discussion on the limitations of the current methods in terms of bacterial mobility constraints and maximum cell number we could quantify. On suggestion, we have added further literature references which provide more context to where our study contributes most and provide comparison with other state-of-the-art microfluidics platforms.

We have performed additional experiments to support our findings: we provide new control experiments including CFU/OD calibration for both strains, batch culture comparisons, phage screens, and interactions with MSSA476 and P278 phage in bulk liquid cultures. We have added a further study on phage-induced morphological changes to PA strain and have shown that we can quantify round-shaped PA cell during the lysis process. This further adds weight to the use of morphological detectors for studying bacteria-phage interactions. In addition, we have clarified many aspects of the work including calculations for cell doubling times, burst size, number of individual droplets screened. Changes to the manuscript as suggested have been addressed and documented in the tracked version of the manuscript.

Overall, we have carefully addressed each point raised by the reviewers in the accompanying point-by-point response and believe that the revised version is now suitable for publication in *Communications Biology*.

Thank you again for your thoughtful review and consideration.

Sincerely,

Fabrice Gielen, Ph.D.

Senior Lecturer

University of Exeter

Reviewer #1

This paper describes a method to analyse the growth of bacteria cultures using droplet microfluidics and an AI-based autofocus function together with morphology-based deep learning models. The experimental system itself has already been published before by the authors (doi:10.3389/fnlct.2023.1258155), however, they extended that with a nice and useful method to study and analyse the growth of polymicrobial communities. The techniques interesting, and could be of interest for the microbiologist community. A limitation of the method is that only the interaction of bacterial strains possessing different morphologies can be studied.

The method itself is promising and could be of interest in the research field, however I have some concerns addressed to the authors. Those should be clarified to see what are the limitations and how robust the system is. Also, they are needed in order to be able to reproduce the experiments.

Major comments:

•My major concern about the manuscript is that the CFU and MOI values are not consistent. In the first part of the paper the authors describe the growth dynamics of two different bacterial species within the droplets. The OD values are summerized in Table 1. However, the authors should also give the CFU values. They use two different species, having different morphologies. The same OD value for *P. aeruginosa* might result in different cell number compared to *S. aureus*. This actually could also be seen in the example presented in Figure 4, both on the images and the graphs presenting the average cell numbers. I wonder if this could be the reason for the longer lag phase of PA14 compared to MSSA476.

Regarding the relationship between OD₆₀₀ and CFU/mL values, we agree that the same OD₆₀₀ value results in different cell numbers for different species. We have now added corresponding CFU/mL values for both cell types in the updated Table 1 of the main manuscript.

We have performed calibration for colony forming units per ml versus optical density measurements using our Clariostar plate reader for both species. All the data were obtained the day of the experiments by plating appropriate cell dilutions on LB agar plates. We note that there is, as expected, a roughly linear correlation between CFU/mL and OD₆₀₀ for the OD₆₀₀ range we tested. We have pooled all the OD₆₀₀ versus CFU/mL measurements in a single graph to get an accurate linear fit. This is now included in Supplementary Information Figures S1 and S2 for both species.

In Table 2. The authors summerize the conditions of the lysis experiments. Here The MOI values are not clear. When they perform lysis experiments with PA14 strain, they claim to use MOI 20, 0.6 and 0.2. However, I can see where MOI=20 comes from, but based on the cell numbers they give, the MOI value should be around 13 and 15 for the other 2 cases. So how did the authors calculated those values? It is a crucial part of the paper. Also, When

they studied the interaction of SA with the phage, they say MOI=0.2 was applied, however, they do not give the CFU number for that strain. I have the same problem with the polymicrobial experiments. CFU-s are not given for the SA strain, and the calculated MOI values are not consistent with the ones above. This part should be clarified within the paper.

MOI values were obtained by mixing specific volumes of bacteria with known CFU/mL with phage titer with known PFU. We have added the exact volume of either cells or phages used in every experiment as listed in the updated Table 2, main manuscript. We have not included MOI value for the SA+P278 phage experiment since MOI is defined for a susceptible strain.

•The authors present only the first couple of hours of the experiments, however they say that they tracked the growth of the culture for about 20 hours. To get proper growth curves, it would be nice to see how the cell number changes in long term. Actually, this would give evidence on how many cells could be analyzed with this method.

We chose incubation times based on observed growth in standard assays which started to enter stationary phase within 4 hours as shown in the Figure R1-1 below.

Figure R1-1. PA growth in 200 μ L batch culture in microtiter plates with 10 biological repeats.

Second, we purposely interrupted data acquisition for growth experiments because it became difficult to obtain accurate counts once droplet capacity was reached. We have clarified (c.f. lines 611-616) that the method is currently limited to detection of \sim 80 cells per droplet based on the current disk exclusion method (c.f. Figure R1-2). This however corresponds to the stationary phase of bacterial growth with an estimated corresponding OD₆₀₀ of well >1 (beyond the linear range of the CFU/mL OD₆₀₀ calibration we have performed). In addition, the droplet growth dynamics reflected well the standard microtiter batch culture in terms of cell doubling time and time to reach the stationary phase.

Figure R1-2: (a) Detections of PA14 cells loaded with high OD₆₀₀ to characterize high bacterial densities, reaching the limits of our counting methodology with the likelihood of two detections across the Z-stack sharing similar xy coordinates within 10 pixels becoming very high (here with over 100 cells detected when the same cells from different slices were found only approx. 1.5µm apart and counted multiple times). **(b)** At very high densities, the model fails to identify all the cells due to overcrowding. **(c)** Standard deviation of error in cell count as the number of cells in the droplet increase.

From our previous study in similar droplet volumes, we showed that error in counting using the current image analysis methodology has a relative standard deviation of 10% above 40 cells¹. The long-term experiments of over 10 hours were more suitable to phage experiments in which there is a more complex interplay between cell division and cell lysis leading to long-term growth suppression assays as seen in Figure 5 in the main manuscript.

•It would be important to clarify how many individual experiments were performed by the authors for each scenarios. Sometimes it is written how many droplets were analyzed, but not all the time. Authors should write how many droplets and how many biological replicates were performed. Also, n should be indicated on the graphs or in the captions

Thank you for pointing this out. The number of individual biological replicates is indicated in Tables 1 and 2. Within each experiment, we analysed up to 20 droplets. The individual numbers are now mentioned in corresponding figure captions (Figures 4, 5 and 6).

•In section 1.7 “Cell doubling timw calculation:

In this section authors should write down in more detailes how they calculated the doubling time. It is not clear how the slope was calculated, which part of the curves were used, also how the authors define growth rate.

As explained in section 1.7, we calculated doubling times from the growth curves. The cell counts with time points were saved in a CSV file and input into a MATLAB script that calculated the fastest specific growth rate, thereby calculating cell doubling time using equation 2.

Specifically, we identified the portions of the cell count plots in log scale with the steepest slopes within a 20 minutes moving window. Examples of where this window was found are given in in SI section Figure S7 and S8. The MATLAB script we used for this purpose is now included on the GitHub page (<https://github.com/GielenLab/YOLOv5-v8-polymicrobial>).

•How did the authors calcualted the burst size? They say that the burst size was calcualted to be 46, but there is no evidence how this was determined.

We now clarify that the burst size was calculated from the rise in PFU/mL during the exponential increase in a one-step growth curve assay (SI, section 5, Figures S10 and S11). It corresponds to the ratio between the end point PFU/mL (plateau phase) and the initial PFU/mL (latent phase) ².

Figure R1-3. Extraction of burst size from the P278 one-step growth curve calculated by dividing the average pfu/mL after the rise period by the average pfu/mL during the latent phase.

Taking the PFU values listed in Table S3, we get the calculation as:

$$\text{P278 burst size} = \frac{PFU_{end}}{PFU_{initial}} = \frac{3 \cdot 10^5}{6.6 \cdot 10^3} \sim 45$$

We added a plot of PFU vs. time in Supplementary Figure S12, derived from Table S3 data, showing the lytic burst and time to lysis (~37 mins)

•**The used mutant strain has a reduced motility; this can also be seen in the videos. It shows mostly Brownian motion. What does happen if we would like to study a swimming strain? I have the feeling that this method cannot be used in such cases. Also, what happens when the culture grows? Is there a maximum number of cells above which the tracking method does not work? How large (how many cells) is the biggest cell culture that can be analysed with this method? Authors should add the limitations of this method into**

the discussion of the manuscript. Adding the growth curves for prolonged periods, not just for 3-5 hours, would also help in seeing the efficiency of this method.

This is an important point. We have now added a paragraph to the discussion acknowledging the limitations of the current method (lines 608-614): it performs best with low-motility and/or non-motile strains. Highly motile bacteria can be detected multiple times across Z-stacks with our current time acquisition being 1.5s for 10 slices. We therefore estimate the maximum linear swimming speed to be $\sim 15 \mu\text{m/s}$ based on our exclusion radius method (3 μm maximum displacement with 0.5 μm interslice spacing). Our exposure time for transmitted light imaging is 19 μs (line 235) which means we do not expect cell appearance to depend on their speed. The method is currently limited to ~ 80 cells per droplet but could be increased with other cell clustering methods making use of the Z positions, or faster Z scanning with piezo mounted objectives or stages (lines 708).

Minor comments:

•There are figures and tables that are not referred in the text (e.g., Table 2, Table5, Figure 2). Also, figures and tables should be positioned after they are first referred in the text. Author should go through the paper and check that all figures and tables should be referred. This is also true for the content of the supplementary informations.

We have corrected this.

•The figure numbers in the supplementary infromations are shifted. Figure S9 is referred as Figure S8 in section 2.1.6, and figure S7 is referred as Figure S6 in section 2.2.

We have checked all Figure numbers for the main and SI files.

•The authors should use more references. There are claims in the introduction that are not supported by any references. E.g., “For example, rapid resistance acquisition and high fitness of resistant bacteria have often been reported when using monocultures.” The authors only have 36 references, so they could add some more to support such statements.

We have added background literature with other state-of-the-art microfluidic platforms as well as additional references for the methods, e.g. CFU/mL calculations, burst size... We also included extended discussions with more references. We believe this additional cited literature (10 further papers) further support our data interpretation and conclusions.

•In the methods section author should define “overnight culture”. How many hours did they culture the cells for the overnight cultures? It could be important when setting the initial cell number. Since they talk about the interaction of bacteria and bacteriophage, it is important to reduce the possible number of resistant bacteria in the initial culture, therefore the duration of the overnight culture and its dilution could be important in these experiments.

The overnight bacterial culture refers to isogenic cultures from a streak left in a shaker incubator at 37 degrees 200rpm for at least 14 hours. In the absence of phages which were

added just before droplets were generated, we do not expect resistance to occur. The following day, we diluted the liquid cultures to a desired OD₆₀₀ and most of the experiments were conducted once the diluted overnight cultures reached OD₆₀₀ values corresponding to exponential phase of bacterial growth. We now clarify this in the main manuscript (line 130).

•In section 1.4, first paragraph: authors use the term dynamic environment, however the environment that the cells encounter with can not be changed throughout the experiment. Therefore, this term is misleading.

We have deleted the term 'dynamic' to eliminate possible confusion (line 224).

•In section 1.5, fourth paragraph: authors refer to “Figure3-ii”, but I think they should use Figure 3A-ii.

This has been corrected.

•In section 2.1.2, first paragraph: Authors write “The flgK mutants had low motility...” In this sentence they should write flgK in italic or with a capital F (FlgK).

This has been changed everywhere as either abbreviated 'PA' or PA14 *ΔflgK*.

•In section 2.1.2: There is no need of Table 4, those results can be summarized in one sentence.

We have removed it.

•I do not understand the Figure S5 in SI file. There is no description of it.

The figure (now Figure S10) is now described more clearly to explain that it was part of a phage library screen which we used to select the phage named P278.

Reviewer #2

In this manuscript, the Authors present a method of image analysis based on machine learning algorithms where the algorithms are trained on morphologies of two bacterial species, one rod and one coccus. In addition, the Authors present an algorithm that autofocuses the acquired images. The Authors show how they employ they methods to image phage infections of bacteria in droplets.

This is a proof-of-principle technology development paper with added validation of the methods presented and does not show any new interesting results in terms of phage-bacteria interactions. I think that the presented image analysis pipeline might be useful for droplet-based assays but is quite not there yet. The Authors use droplets, but they only use 20 droplets in an assay – it does not make sense to involve droplets at this scale. The Authors mention that the “traditional” assays require a single layer of cells to work, but their droplets are essentially flat (6 µm high). The autofocus algorithm that the Authors present seems to be doubling the function of existing hardware-based microscope modules (such as Nikon’s Perfect Focus or Leica’s Adaptive Focus Control) or even software-based autofocus modules, thus I am unsure of its novelty – albeit it might be useful in low-resource settings.

Even though the present study is small scale, it enables time-lapse characterization of small, independent bacterial communities imaged at the single-cell level. The droplet format is helpful to generate initial diversity and obtain biological repeats with slight differences in the initial number of cells and phages based on Poisson loading. The diversity of growth profiles in co-culture experiments is largely due to these differences in initial cell densities. In addition, monitoring droplets loaded with various numbers inform, as we note, on the inoculum effect, i.e. the influence of initial cell density on response to phage exposure. On top of what is presented, this study sets the scene for higher throughput formats in which droplets are imaged in-flow.

Many autofocus modules have been developed for various purposes, but we found that standard methods such as histogram-based autofocus failed at keeping consistent focus over time, and non-aqueous features of microfluidic devices can interfere with hardware-based autofocus. In addition, proprietary autofocus functions from microscope manufacturers are typically not provided as free open-source codes. The autofocus method developed was found to be effective at ensuring consistent focus where other methods failed. This exemplifies a further utility of image-based deep learning for the purpose of long term focus management.

Some specific comments:

- 1) A major issue: Why did the Authors use Δ flgK mutant for *P. aeruginosa*? They say it is because it has impaired mobility in comparison to wild type and is thus easier to run image analysis on it. What are the limitations of the presented image analysis algorithms in terms of motility of analysed bacteria? The Authors account for**

bacteria movement between slices in chapter 1.6. Have the Authors prepared an analysis of bacterial movements at a high frame rate of imaging to establish that 3 μm diameter sphere is a good basis to assume that a bacterium will not move more than this distance between frames? Does it mean that the highest number of cells analysed in a droplet is determined by the number of virtual spheres around cells, each sphere 3 μm in diameter?

We chose the PA14 ΔflgK mutant for its reduced motility, which enables accurate counting of cells. Highly motile bacteria such as wild-type PA14 (average swimming motility in the range of 50 microns per second³) would result in multiple detections of the same cell across Z-stacks. With our current time acquisition being 1.5 seconds for 10 slices, we estimate the maximum linear swimming speed of a chosen bacteria to be $\sim 15 \mu\text{m/s}$ based on the exclusion sphere method (3 μm maximum displacement with 0.5 μm interslice spacing). We have added a discussion on this limitation in the manuscript (lines 611-617).

To assess movement thresholds, we analysed high-frame-rate time-lapse images and found that bacterial displacements for PA14 ΔflgK across Z-slices were within a 3 μm sphere for a 1.5 second interval (exemplified by the multiple detections for the same cells shown in Figure R2-1).

Figure R2-1. Example image with cells detections. Green dots clustering around a red dot represent detections for the same cell across different Z slices. Red dots represent individual cells filtered by the exclusion sphere method, ensuring each cell is counted only once.

This formed the basis for our sphere of exclusion across Z-stacks to count each cell only once across several planes. The method is currently limited to ~ 80 cells per droplet (based on sphere packing) but could be increased with more efficient 3D clustering methods. We used a distance metrics to determine if detections from different Z-slices belong to the same cell but could also use more advanced clustering algorithms such as DBSCAN (Density-Based Spatial Clustering of Applications with Noise) or hierarchical clustering to group the detections.

To improve on both the motility range and the total number of countable cells, the speed at which the focal depth is scanned must be improved, e.g. making use of piezoelectric Z-stages, piezo-objective scanners or electrotuneable lenses (this comment was added line 706).

2) How would the system perform if a droplet was not flat but spherical of the same volume as the Authors used, 11 pl?

Spherical droplets of same volume (~28 μm diameter) would require ~56 Z-slices for full scanning, which in turn would require 7.5 seconds for image acquisition, significantly increasing the time and size of datasets required to image multiple droplets. Our current setup and detection method is currently optimised for quasi-flat droplets (6 μm in height). We have added a note in the revised results, line 385.

‘The quasi-flat droplets (6 μm in height, 60 μm in diameter) decreased the need for large numbers of Z slices, significantly speeding up the overall image acquisition process while still allowing cells to swim unconstrained.’

3) The Authors compare their system to traditional methods of analysis of polymicrobial cultures. Can the Authors prepare a similar brief comparison to state-of-the-art microfluidic systems so that the reader has a better oversight on the novelty of the presented method? There are labs of Oscar Hallatchek, Jeff Hasty, Howard Stone, Bartłomiej Waclaw, late Alma dal Co, Simon van Vliet...

Recent innovations in microfluidics have enabled powerful approaches to study microbial communities. Within single phase devices, the mother machine devices have been extensively used. For instance, Alma Dal Co and Simon van Vliet have advanced the field with devices to study ecological interactions and feedback between environmental structure and microbial behaviour, though most systems rely on flow-through or 2D culturing formats. The Hallatschek’s and Waclaw’s groups have focused on spatial dynamics and stochastic effects in bacterial colonies using planar microfluidic chambers and agarose-based devices, highlighting genetic drift and sectoring phenomena. Early droplet-based co-cultivation systems such as that of Herrera-Estrella *et al.* (2011) demonstrated the utility of fluorescently labelled bacteria for high-throughput mapping of microbial interactions in droplets, highlighting the potential of droplet platforms for studying microbial consortia.

There have also been innovations in droplet-based screening methods: the Hasty’s lab has pioneered synthetic gene circuits and quorum sensing systems within droplet-based setups, often requiring fluorescent reporters.

In contrast, our approach uniquely combines long-term confinement of polymicrobial co-cultures in picolitre droplets with label-free time-lapse Z-stack imaging, automated deep-learning-based autofocus, and morphology-based species-specific detection using YOLOv5. Unlike fluorescence-reliant studies (e.g., co-encapsulated symbiotic bacteria), our method preserves native cell genetics and avoids labelling bias, enabling continuous quantification of growth and phage-induced lysis in mixed communities. This expands the applicability of microfluidic studies towards label-free infection dynamics in complex communities. A summary table listing the state-of-the-art microfluidics methods is given below and has been added to SI section and Table S4.

Lab / Study	Microfluidic Format	Cell Detection Method	Polymicrobial Support	Phage Interaction Studied	Fluorescence Requirement	Quantitative Output
This Study	Anchored picolitre droplet arrays	Label-free deep learning (YOLOv5, morphology-based)	Yes (e.g., PA14 and MSSA476)	Yes (quantitative phage-bacteria dynamics in droplets)	No (fully label-free detection)	Growth curves, doubling time, lysis dynamics
Hallatschek (2019), ⁴	Planar microfluidic chambers / agarose devices	Imaging / fluorescent reporters	Yes (coexistence, spatial segregation)	No or minimal (focus on population dynamics without phages)	Yes (often required for tracking)	Growth patterns, spatial segregation
Hasty et al. (2009), ⁵	Droplets / microchambers with synthetic circuits	Fluorescent reporters (synthetic constructs)	Limited (focus on engineered E. coli strains)	Rarely (focus more on genetic circuits)	Yes (integrated with circuits)	Circuit dynamics, population trends
Ramachandran et al. , 2024, ⁶	Flow-based microchannels and trapping devices	Brightfield / fluorescent microscopy	Limited (mostly monocultures or passive transport studies)	No	Often yes	Transport, mixing, shear impact
Dal Co et al. (2019, 2020), ^{7,8}	Multilayer PDMS devices for ecological studies	Brightfield, phase contrast, fluorescence	Yes (spatially structured communities)	No	Yes (common)	Spatial structure, ecological stability
Park et al. (2011) ⁹	Droplet-based co-cultivation in microwell array	Fluorescent imaging	Yes (symbiotic interactions in co-encapsulated)	No	Yes (fluorescent reporters used)	Co-culture viability and interaction network mapping

			communities)			
--	--	--	--------------	--	--	--

- 4) The Authors say that “most” current imaging methods require cells to grow in single layers – what are the current imaging methods that do not have this requirement? I think about the research done in 3D in the groups of Knut Drescher (<https://www.nature.com/articles/s41564-020-00817-4>), but there may be others?**

We thank the Reviewer for highlighting this point. Indeed, while many high-throughput or quantitative imaging methods rely on monolayers for ease of segmentation and quantification—such as widefield or high-content screening platforms—there are notable exceptions that explore 3D bacterial environments. For example, the work by Knut Drescher’s group (e.g., Hartmann et al., Nat Microbiology 2021) employs light-sheet fluorescence microscopy (LSFM) to study bacterial biofilms in 3D, enabling spatially resolved single-cell tracking over time. Similarly, work by Joshua Shaevitz, Bonnie Bassler, and others have used confocal or spinning disk microscopy to study 3D aggregates and biofilm formation, though often in fluorescently labelled strains. However, these methods are only applicable to immobilised cells and therefore cannot be easily adapted in the context of planktonic cell analyses.

Our use of brightfield Z-stack imaging in anchored droplets represents a complementary approach where unlabelled cells can be quantified over time in confined 3D geometries. While LSFM and confocal approaches offer unmatched resolution in 3D volumes, they typically require transparent media, optical access from multiple angles, and complex instrumentation. In contrast, our platform offers a compact, automated and label-free method for studying polymicrobial dynamics and phage-mediated lysis in bacterial populations within droplets. We have added this discussion to the introduction of the main manuscript (line 91-92).

- 5) I agree that quantification of motile cells in droplets is more challenging technically that of cells in biofilms or in single layers. The Authors, however, do not make a strong case about why it is important to be able to study cells in liquid phase instead of on surfaces, especially that they do not perform their later described experiments at a large scale in droplets (they use up to 20 droplets), thus not utilizing the possibility of screening a large parameter space which would be a major advantage of droplet use over traditional methods.**

Studies have shown that formats (cell layers, biofilms, planktonic) in which bacterial cells grow influence their gene expression and in turn their response to antimicrobials. In a seminal study, Hoiby and colleagues showed that the formation of *P. aeruginosa* inhibition zone during tobramycin agar diffusion susceptibility tests is due to a switch from planktonic growing bacteria to the biofilm mode of growth¹⁰. Recently, Ramachandran *et al.* showed that planktonic cells exposed to shear flow have altered transcriptomics profiles⁶. The study of the

link between cell motility and virulence is also important as in clinic contexts including respiratory mucus environments (e.g., cystic fibrosis), and wound exudates ¹¹. Furthermore, complex 3D environments, such as those with porous structures, can influence bacterial colonization patterns and quorum sensing.

We now better explain these essential differences to further motivate the choice of the droplet format (lines 74-81).

While our current imaging experiments were limited to ~20 droplets due to the need for high temporal and spatial resolution Z-stack acquisition, the underlying platform is fully scalable using faster automation instruments. The anchored droplet chip design supports hundreds of droplets, and the automation scripts are compatible with expanded stage scanning. Future use of parallel imaging setups, reduced Z-stack depth, or selective timepoints will enable higher-throughput screening of parameter spaces. Discussion has been added line 726.

6) Microbial interactions have, indeed, been studied in droplets, also label free and at a large scale: Kehe et al. <https://www.pnas.org/doi/10.1073/pnas.1900102116>

We thank the Reviewer for pointing out the important contribution by Kehe et al. (PNAS, 2019), which used fluorescent color codes to avoid the use of fluorescent strains and fluorescent-based assays. In this study using nanoliter droplets, the authors infer microbial interaction networks at scale but only study population level effects. Our platform differs in key ways: we implement brightfield Z-stack imaging combined with morphology-based deep learning to identify and count individual cells of two species over time, rather than relying on bulk fluorescence assays. This allows us to track species-specific dynamics, detect cell lysis events, and analyze polymicrobial-phage interactions with single cell resolution. While Kehe et al.'s approach excels in scale and interaction mapping, our method enables insights into morphological changes during phage-induced lysis or the quantitative study of competitive growth between species. We now cite Kehe et al. and their approach for studying microbial communities (line 70).

7) I admire the Authors' honesty about using scotch tape to immobilize microscopy samples. I use scotch tape and blu tack.

8) Why is detection of PA14 more reliable than of MSSA476 as per Table 4?

We thank the Reviewer for raising this point. The higher detection performance for PA14 (mAP_{0.5} of 99.4%) compared to MSSA476 (90.5%) likely arises from differences in morphologies within each species: PA14 cells exhibit a well-defined, elongated rod-shaped morphology. In contrast, MSSA476 cells are cocci that often appear as singlets, diplococci, tetrads or grape-like aggregates, and can exhibit irregular or overlapping forms. This morphological variability likely reduces object detection accuracy. This discussion has been added lines 422-425.

9) Why there are seemingly PA14 bacterial cells that survive (not a rare event, as in Fig. 5B the mean number of surviving bacteria even including standard deviation is above 0) MOI 20 of P278 as in Fig. 5?

Residual detection of rod-shaped bacteria were observed in many droplets. However, the lack of growth long-term indicates that such cells, although displaying correct morphology, may not be viable (c.f. lines 532-534).

False positive detections could also occur: The YOLOv5 model occasionally detects morphological debris (e.g., ghost cells, lysed fragments) as intact PA14 cells—especially if their elongated shape remains partially preserved. These false positives contribute to non-zero residual cell counts. We implemented filters (e.g., static detection removal), but they cannot fully eliminate such artifacts.

A second hypothesis is phage loading stochasticity : although MOI is calculated as an average, stochastic phage loading into droplets means some droplets may initially contain slightly lower phage-to-cell ratios. However, in MOI 20 experiments, we did not observe any droplets with sustained PA14 growth, suggesting most cells were effectively lysed and the remaining non-viable.

We also observed that most PA14 cells formed spheroplast-like structures during the lysis process. These cell forms may retain phage particles internally and delay lysis of other cells (SI section 9).

We have clarified this point in the revised text (line 622-632) and added a note that residual cell counts may reflect non-viable remnants or delayed lysis rather than true survival or resistance.

Figure R2-2. Example PA cell with rod-shape morphology remaining after 25 min at high MOI. Round-shaped PA cells in the process of lysis are also visible.

10) Why is the number of analysed droplets 6, 10, 20? Why does the method not support high throughput image analysis of thousands of droplets? This raises questions about the claim in discussion that the presented method presents “large datasets for both technical and biological repeats”.

We thank the Reviewer for this important point. We clarify that although each experiment includes 6–20 droplets, each droplet is imaged using high-resolution Z-stacks at frequent time intervals (e.g., 40 Z-slices every 5 minutes for up to 24 hours). This results in excess of 20,000 images per droplet, yielding over 200,000 images per experiment, which are processed into millions of detection events across multiple timepoints and conditions. We agree that this constitutes rich image data, rather than high droplet throughput per se. We have revised the manuscript text to clarify that our “large dataset” refers to deep, high-resolution time-lapse datasets, not large-scale droplet screening (lines 722-726).

11) The Authors say that by targeting an average of 10 cells per droplet, some cells contained as little as one bacterium. By using Poisson distribution at expected value of 1 and mean of 10, we can see that 0.45% of droplets should contain one bacterium when the mean was 10. As the Authors use 20 droplets per experimental condition at most, it is unlikely that any of their droplet contained a single bacterium.

We agree that at an average loading of 10 cells per droplet, the probability of a droplet containing exactly one cell is indeed very low (~0.45%). To clarify: single-cell droplets were only relevant in our early growth experiments, where we deliberately targeted a lower mean cell number (e.g., 1–3 cells per droplet) by adjusting OD₆₀₀ during loading. In contrast, for phage experiments, we targeted a mean of ~10 initial cells per droplet to ensure sufficient population size for monitoring lysis kinetics. We have revised the manuscript to distinguish clearly between the droplet loading strategies for growth vs. phage experiments and to avoid misinterpretation regarding single-cell occupancy at high mean loading values (lines 434).

12) Inoculum effect of antibiotics on bacteria has already been studied in droplets, it is fair to cite as no explanation of inoculum effect by the Authors is introduced. Postek et al. <https://pubs.rsc.org/en/content/articlelanding/2018/lc/c8lc00916c>

We agree that the work by Postek et al. (Lab Chip, 2018) is an important study demonstrating the inoculum effect in droplet microfluidics using antibiotic gradients. While our focus was on phage-bacterial interactions, not antibiotic exposure, we also observe cell-density-dependent lysis dynamics, which are conceptually related. We have now cited this study in the discussion and clarified that inoculum effects have been previously characterized in droplet systems, though our work applies this concept in a label-free, phage-specific context (line 694).

Reviewer #3

The authors co-cultured *Pseudomonas aeruginosa* and *Staphylococcus aureus* inside a single droplet and monitored their relative growth kinetics for up to 24 hours on an imaging basis with deep learning. They challenged *Pseudomonas aeruginosa* lytic phages with monocultures and multicultures and quantified the growth rates, relative bacterial density, and lytic kinetics of the two species. The content is very interesting and unique. I would like to suggest that this article will be published in *Communications Biology* after major revision. The author should address followed concerns.

(1) As a basic rule, when data or schematic diagrams presented in Figures are described in the text, it should be stated exactly which Figure they correspond to. It will aid the reader's understanding. Please correct the following :

- **Which sentence applies to the explanation of Figure 2, starting around L211? Correct it.**

This has been done (line 244).

- **In L255, is Figure 3-i a mistake for Figure 3A-i? Please state exactly. Similarly, in L261, Figure 3-ii should also be stated correctly.**

This has been changed.

- **Which sentence corresponds to Figure 3B and Figure 3C? Correct each.**

This has been added (line 333 and 336).

- **Which sentence corresponds to Figure 4C? It probably corresponds to Section 2.1.6. Correct it.**

This has been added (line 463).

- **Which sentence corresponds to Figure 6B? It probably corresponds to Section 2.2.3. Correct it.**

This has been added (line 556).

(2) I got the impression that the discussion after paragraph starting from L537 is lacking. In particular, the discussion of the series of microbial growth in the droplet is poor. Please respond to the following points.

- **L541: These findings highlight intrinsic growth rate differences in the droplet format, likely influenced by metabolic disparities and environmental adaptation.**

The authors cite metabolic disparities and environmental adaptation as reasons for the 20-minute slower SA doubling time in droplets than in bulk culture. Of course, since microorganisms are confined to the microscopic space of a droplet, adaptation to the environment and the metabolic changes that result from it would be well taken into

account. However, the authors' experiment is the same in that they used a rich medium, which is considered to be an adequate environment for microbial growth. On the other hand, are differences in culture methods considered as a factor? For example, in reference No. 34, they use concussion culture. The concussion culture has the effect of aeration. Regarding aeration, in this experimental system, sufficient oxygen concentration is provided, as the authors describe that the oil phase dissolves 10 times more oxygen than water. Concussion culture also has a role in physically aiding cell division. Droplet cultures are static cultures, which eliminates the physical effects of shaking, but do you consider the possibility that this may have slowed down the cell doubling time?

Indeed, while we initially attributed the longer doubling time of *S. aureus* in droplets to metabolic disparities and environmental adaptation, we agree that mechanical differences in culture formats—especially static versus shaking conditions—likely contribute significantly.

In our experimental system, droplets are immobilized and thus lack the physical agitation provided by shaking or "concussion" cultures, such as those used in Ref. 34. As the reviewer notes, shaking not only improves oxygenation but also provides mechanical stimuli that can assist in nutrient distribution and physical separation during cell division, which has been shown to be important for *S.aureus* for which cells do not easily separate from parents cells.¹²

We have already discussed oxygen availability in our setup, where the fluorinated oil phase can dissolve ~10× more oxygen than water, which helps mitigate oxygen limitation. However, we now also acknowledge that the absence of mechanical mixing in droplet confinement may effect our count estimate as SA cells form clusters in which individual cells cannot be easily identified.

We have added this clarification to the discussion (line 650-652) and expanded the interpretation to include physical confinement and static culture effects as contributing factors to the observed growth rate differences between droplet and bulk cultures.

• L546: However, MSSA476 sometimes outcompeted PA14 (i.e. in approximately 10% of our experiments, Supplementary Figure S9). Please add which droplet in Figure S9 corresponds to this phenomenon.

We recognise that the term "outcompete" refers broadly to the ability of one bacterial strain or species to dominate over another and this can be due to several factors including but not limited to growth rate and cell densities. Here we consider that SA outcompeted PA if cell numbers were higher at the end point of 200 minutes for our experiment, independently of initial PA:SA cell numbers/ratios. In the example of droplet 12 (new Figure S14), there were fewer SA cells than PA cells initially but there were more after 200 minutes incubation. The co-culture dynamics appear to be complex and capturing this long-term growth trends would require longer acquisition times. However, by the end of the experiments (200 min), most

droplets contained over 50 cells in total, which approached the counting limit with our current method.

We have updated the legend of Figure S14 to mark droplets where *S. aureus* outnumbered *P. aeruginosa* at later time points. Specifically, droplets 8 and 12 showed this trend. This has now been stated explicitly in both the Results (line 657) and SI captions Figure S14.

• **L547-54: the droplet format is more conducive to balanced growth for the two species compared to traditional formats.**

Even though SA doubling times are different from bulk cultures, Why are droplet cultures better than bulk cultures? The authors state that this is in comparison to bulk culture, but are they also co-culturing bulk in this study? If the authors are doing experiments, shouldn't they present that data for discussion? Or are they trying to argue that they are better suited to looking at the growth of individual microorganisms as opposed to bulk culture? It would be better to state exactly what you mean, as it could be confusing to the reader.

Our intent was not to suggest that droplet cultures promote faster or more “balanced” growth overall, but rather that the droplet format enables species-resolved quantification of growth dynamics in polymicrobial systems, which is difficult to achieve in bulk cultures. In bulk co-cultures, optical density (OD₆₀₀) cannot differentiate between species, and plating on selective media is both time-consuming and destructive.

To address the reviewer’s suggestion, we have now conducted bulk co-culture experiments of *P. aeruginosa* PA14 Δ flgK and *S. aureus* MSSA476 under the same LB medium and incubation conditions (new Figure S9, SI section 4). These data show that while PA grows comparably in both mono- and co-cultures, SA reaches significantly lower densities in polycultures, with its population reduced by over an order of magnitude after 24 hours. This outcome aligns with previous literature reporting *S. aureus* suppression by *P. aeruginosa* in bulk formats¹³. In contrast, the droplet format revealed a subset (~10%) of droplets where MSSA476 was able to outcompete PA14, suggesting the stochastic encapsulation, physical isolation and overall droplet format led to higher heterogeneities.

• **L564: We hypothesize that these extended cell doubling rates could be due to the presence of PA14 cell lysate and high numbers of phage particles within each droplet.**

Why would a large amount of phage particles increase the cell doubling time? Assuming that, we can examine this by culturing MSSA476 together with a large amount of phage in droplets, as shown in Figure 5d. Why has that experiment not been performed?

Furthermore, in the absence of competitor P14, would MSSA476 not be inhibited in growth, allowing it to grow at the same doubling time under the same conditions as when cultured monoculture? For example, if P14 and MSSA476 are co-cultured under MOI 20 conditions, it is assumed that P14 is immediately lysed and the phage itself cannot be increased. In this case, is the doubling time of MSSA476 normal? Co-culturing P14 and MSSA476 under

different conditions, such as MOI 20 or 0.6 in monoculture, would help further discussion on the effect of the number of phage particles present on the doubling time of MSSA476.

Are there any other possibilities? For example, since it takes a lot of energy for phage to increase within P14, many media components are utilized, reducing the media components available for MSSA476 to proliferate. A disadvantage of droplet culture is that the droplet volume is pL scale, so it is quite possible that nutrients will become depleted, and the growth will stop. I think it would help researchers who deal with microorganisms if we discuss this point of view as well.

While we initially tested *S. aureus* growth with phage P278 at MOI 0.2 (Figure 5D) and found no impact on growth rates, we agree this did not rule out possible indirect effects at higher phage levels in co-culture.

To address this, we performed a control experiment in a microtiter plate format, growing MSSA476 alone in LB medium in the presence of high P278 titres (at equivalent MOI 5). The resulting growth curves, now presented in Figure S17, confirm that P278 phage alone has no direct inhibitory effect on MSSA476, even at high MOI.

This finding suggests that the delayed growth observed in PA14+MSSA476+phage droplets is likely due to indirect effects of phage infection and PA14 lysis, rather than direct phage toxicity.

We have now elaborated on several plausible mechanisms that could contribute to the increased doubling times observed of MSSA476 in co-culture with phage conditions:

Cell underestimation (lines 650-652): as MSSA476 divides in absence of mechanical shear forces, the cells do not easily separate from parents cells and form aggregates in which individual cells are difficult to count, leading to actual numbers of individual MSSA476 being higher than detected clusters.

Nutrient depletion (line 679): As the reviewer rightly notes, high levels of phage replication within PA14 could rapidly deplete local nutrient pools (e.g. nitrogen) in the picolitre-scale droplets, leaving less available for MSSA476. Other hypotheses could be the accumulation of cell debris altering osmolarity or viscosity or the release of toxic virulence factors or degradation products ¹⁴.

We also agree with the reviewer that additional co-culture experiments at different MOIs (e.g., 0.2, 0.6, 20) with constant MSSA476 presence could help disentangle these effects. Due to time constraints, we have not yet performed all these variations, but we have added this proposal as a future direction in the revised discussion.

(3) In conclusion, the authors discuss the potential of this technology but do not discuss the problems and limitations of the current situation.

It would be appropriate to add a discussion of the limitations of distinguishing endemic species from microbial morphology by imaging. Since PA14 and MSSA476 are clearly different in morphology, as in this study, this would have been good material for an initial

POC to demonstrate the authors' concept. However, with the numerous number of microbial species in the natural environment, is there a limit to morphological classification? For example, is it possible to distinguish between different species of microorganisms that are very similar in morphology? In addition, when it comes to distinguishing between three or four microorganisms, the difficulty of identifying microorganisms instantly increases, and it is easy to imagine how difficult it would be to distinguish between them. The authors should be a bit more careful in describing, as it could mislead the reader.

We clearly caution against overgeneralizing the applicability of the current model to more complex natural communities (line 727-734). This study used morphologically distinct strains (rods vs cocci) as a proof of concept. For closely related or similarly shaped species, morphology alone may not suffice, especially given cells have varied orientations in liquid phase. Misclassification rates would likely rise with >2 species unless paired with additional labelling (e.g. fluorescence) or phenotypic signatures. In future work, expanding to instance segmentation, or hybrid fluorescent-morphological approaches could mitigate these challenges.

To demonstrate further use of the morphology-based detection method, we have performed an additional study focused on the transition of PA14 from rod-shape to round cell morphology during the phage lysis process (SI section 9).

We hypothesise that these round 'cells' are spheroplasts because of their characteristic round shape and lower contrast of their membrane following digestion of the peptidoglycan membrane by the phage endolysins (Figure S16). However, further confirmation would be required to confirm spheroplasts such as membrane staining or extraction followed by peptidoglycan quantification. We found that these round cells were not detected in all experiments, indicating that there may be sometimes short lived. In the experiments where we could detect them, the number of spheroplasts-like cells in a droplet approximately matched the number of cells being lysed, indicating that the round morphology is a reproducible intermediate shape during the overall lysis process. An example lysis experiment in which round shape cells are seen is displayed in Figure S16.

Significantly, phage particles may remain encapsulated within these spheroplast-like cells, delaying the release of newly synthesized phages and therefore delaying the lysis of other cells. This observation may help in deciphering lysis dynamics (line 629).

However, when trying to detect spheroplasts-like PA14 cells in a co-culture of PA14 and MSSA, we found that the model could not accurately distinguish round cells from MSS476 due to their round morphologies being too similar.

This additional study confirms that morphology-based object detection is versatile in the type of object under detection as long as the features it encodes are well distinct between the different training classes. It also exemplifies how morphology can be tracked during the lytic process to infer on the stage of infection.

References

- 1 Tiwari, A., Nikolic, N., Anagnostidis, V. & Gielen, F. Label-free analysis of bacterial growth and lysis at the single-cell level using droplet microfluidics and object detection-oriented deep learning. *Frontiers in Lab on a Chip Technologies* **2**, doi:10.3389/frlct.2023.1258155 (2023).
- 2 Kropinski, A. M. Practical Advice on the One-Step Growth Curve. *Methods Mol Biol* **1681**, 41-47, doi:10.1007/978-1-4939-7343-9_3 (2018).
- 3 Son, K. & Stocker, R. Analyzing and Using Motility Kinematics of Microorganisms. (2018).
- 4 Gralka, M. & Hallatschek, O. Environmental heterogeneity can tip the population genetics of range expansions. *Elife* **8**, doi:10.7554/eLife.44359 (2019).
- 5 Bennett, M. R. & Hasty, J. Microfluidic devices for measuring gene network dynamics in single cells. *Nat Rev Genet* **10**, 628-638, doi:10.1038/nrg2625 (2009).
- 6 Ramachandran, A., Stone, H. A. & Gitai, Z. Free-swimming bacteria transcriptionally respond to shear flow. *Proc Natl Acad Sci U S A* **121**, e2406688121, doi:10.1073/pnas.2406688121 (2024).
- 7 Dal Co, A., van Vliet, S., Kiviet, D. J., Schlegel, S. & Ackermann, M. Short-range interactions govern the dynamics and functions of microbial communities. *Nat Ecol Evol* **4**, 366-375, doi:10.1038/s41559-019-1080-2 (2020).
- 8 Dal Co, A., van Vliet, S. & Ackermann, M. Emergent microscale gradients give rise to metabolic cross-feeding and antibiotic tolerance in clonal bacterial populations. *Philos Trans R Soc Lond B Biol Sci* **374**, 20190080, doi:10.1098/rstb.2019.0080 (2019).
- 9 Park, J., Kerner, A., Burns, M. A. & Lin, X. N. Microdroplet-enabled highly parallel co-cultivation of microbial communities. *PLoS One* **6**, e17019, doi:10.1371/journal.pone.0017019 (2011).
- 10 Hoiby, N. *et al.* Formation of *Pseudomonas aeruginosa* inhibition zone during tobramycin disk diffusion is due to transition from planktonic to biofilm mode of growth. *Int J Antimicrob Agents* **53**, 564-573, doi:10.1016/j.ijantimicag.2018.12.015 (2019).
- 11 Higgs, M. G. *et al.* Flagellar motility and the mucus environment influence aggregation-mediated antibiotic tolerance of *Pseudomonas aeruginosa* in chronic lung infection. *mBio* **16**, e0083125, doi:10.1128/mbio.00831-25 (2025).
- 12 Missiakas, D. M. & Schneewind, O. Growth and laboratory maintenance of *Staphylococcus aureus*. *Curr Protoc Microbiol* **Chapter 9**, Unit 9C 1, doi:10.1002/9780471729259.mc09c01s28 (2013).
- 13 Filkins, L. M. *et al.* Coculture of *Staphylococcus aureus* with *Pseudomonas aeruginosa* Drives *S. aureus* towards Fermentative Metabolism and Reduced Viability in a Cystic Fibrosis Model. *J Bacteriol* **197**, 2252-2264, doi:10.1128/JB.00059-15 (2015).
- 14 Shah, R., Jankiewicz, O., Johnson, C., Livingston, B. & Dahl, J. U. *Pseudomonas aeruginosa* kills *Staphylococcus aureus* in a polyphosphate-dependent manner. *bioRxiv*, doi:10.1101/2023.12.05.570291 (2023).

We thank all the reviewers for their second careful reviews of our revised manuscript. We have addressed the suggested changes by Reviewer 1 and incorporated them into a new revised manuscript and supplementary files. We detail our revisions below:

Reviewer #1:

The authors addressed all of my concern, the revised manuscript contains detailed information about the experiments and also the discussion of the manuscript became more prudent.

I have some comments that should be addressed, but after these corrections I suggest the MS to be accepted for publication (minor revision).

1) In the abstract the authors claim that their method is suitable to follow the growth dynamics of *P. aeruginosa* and *S. aureus* for 24 hours. I feel that this is misleading, and it is an overstatement. The authors can only follow the killing curves of the phage for long-term. If cells are growing they mention that there is a limitation in cell number to follow... Also there are not too many graphs presented in the paper with the time scale of 20 hours... Most of the growth curves were followed for 4-5 hours. So, the authors should remove this sentence (“We show that we can monitor the relative growth dynamics of *P. aeruginosa* and *S. aureus* growing in 11 picolitre droplets for up to 24 hours.”) from the abstract

We agree that long-term growth cultures may exceed our detection limit for long time periods and have therefore changed the abstract to highlight this capability can be used for phage experiments in which growth and lysis co-exist.

Line 26: ‘We monitor the interactions between bacterial mono- or co-cultures of *P. aeruginosa* and *S. aureus* in the presence of a *P. aeruginosa* phage growing in 11 picolitre droplets for up to 20 hours’

2) The CFU graph of PA14 mutant strain and the data in Table S1 are not consistent in the Supplementary Informations. Also the values on the y axis...I guess they should be multiplied by 10⁸ or 10⁹ ? And it is also not consistent with the data presented in Table 1 in the manuscript. Please check these data and correct it! For the other bacterial strain (MSSA476) it is ok.

Thank you for pointing this mistake out. We have now corrected the y axis in the updated Figure S1 (multiplied by 10⁹).

3) I see no reason why the last chapter of the Supplementary Informations (Comparison to microfluidics-based studies) is not included in the MS. It would be nice to add such concept into the discussion and/or introduction. Maybe leaving

the table S4 in the supplementary file, but adding the discussion part into the manuscript. Please consider it.

We have added the first paragraph of this discussion from SI into the main MS introduction, starting line 74.